# SABER: Small Actions, Big Errors — Safeguarding Mutating Steps in LLM Agents

## Abstract

Despite rapid progress in LLM agents, performance on long-horizon, tool-using tasks remains fragile. To better understand this fragility, we ask a simple question: *do all actions contribute equally to failure?* Analyzing execution traces on $\tau$-Bench (Airline/Retail) and SWE-Bench Verified, we decompose trajectories into *mutating* (environment-changing) vs. non-mutating steps and formalize *decisive deviations*—earliest action-level divergences that flip success to failure. A logistic regression reveals that each additional deviation in a mutating action reduces the odds of success by upto 92% on Airline and upto 96% on Retail for SoTA models. In contrast, deviations in non-mutating actions have little to no effect. Errors also grow with context length as agents drift from role and act on stale constraints. Motivated by these observations, we introduce SABER, a model-agnostic, gradient-free, test-time safeguard that (i) adds mutation-gated verification, (ii) injects *Targeted Reflection* before mutating steps, and (iii) performs block-based context cleaning. SABER delivers consistent gains—e.g., Qwen3-Thinking: +28% *relative* on Airline, +11% on Retail, and +7% on SWE-Bench Verified; Claude: +9%/+7%. We further identify ceiling effects in $\tau$-Bench, where annotation errors and underspecified tasks artificially cap model performance. To address this, we release $\tau$-Bench Verified, which restores benchmark headroom through targeted revisions. Our results argue for action-level analysis, targeted safeguards, and reliable evaluations as prerequisites for robust multi-turn agents.

## 1 Introduction

Real-world, long-horizon tasks—whether in enterprise operations, software engineering, scientific analysis, or multi-step information retrieval—demand language agents that can plan, invoke tools, and maintain coordinated behavior across many turns (Chen et al., 2025; Kanoulas et al., 2025; Yang et al., 2024). Despite impressive single-step capabilities, today's leading agents are brittle in extended interactions: they misinterpret constraints, rely on stale context, and issue tool calls that derail progress (Jimenez et al., 2024; Yao et al., 2024; Kwan et al., 2024; Wang et al., 2024b). Current frameworks typically treat all decision steps uniformly—end-to-end prompting, generic scoring, and whole-trajectory reruns all assume the same level of scrutiny across actions (Park et al., 2023; Yuan et al., 2025; Chen et al., 2024b; Zhou et al., 2025; Chhikara et al., 2025; Han et al., 2025). Recent analyses catalog broad behavioral failures (Zhang et al., 2025; Cemri et al., 2025), but rarely pinpoint the specific decision steps where success flips to failure. Our study begins with a simple question: *Do all actions contribute equally to task failure?*

We answer this by analyzing execution traces of strong open- and closed-weight models on $\tau$-Bench (Section 3). Partitioning the action space into *mutating* (state-changing such as cancelling a booking, issuing a refund, deleting a file) and *non-mutating* (information-gathering) steps, we show by fitting a regression model that deviations in mutating actions are the decisive predictors of failure, with each additional deviation in the number of mutating actions reducing the odds of success by 55% ∼ 92% on the Airline subset and 87% ∼ 96% on the Retail subset (all $p < 0.001$), respectively, for three different models including Qwen3-Thinking-235B, GPT-5 and Claude-4-Sonnet. Meanwhile, deviations in non-mutating steps have little effect: always below 10% success ratio reduction per non-mutating deviation on both Airline and Retail subsets, with non-significant $p$-values on some cases (details in Table 1). In short, failures cluster at a small slice of mutating steps, revealing a disproportionately risky subset of the decision space.

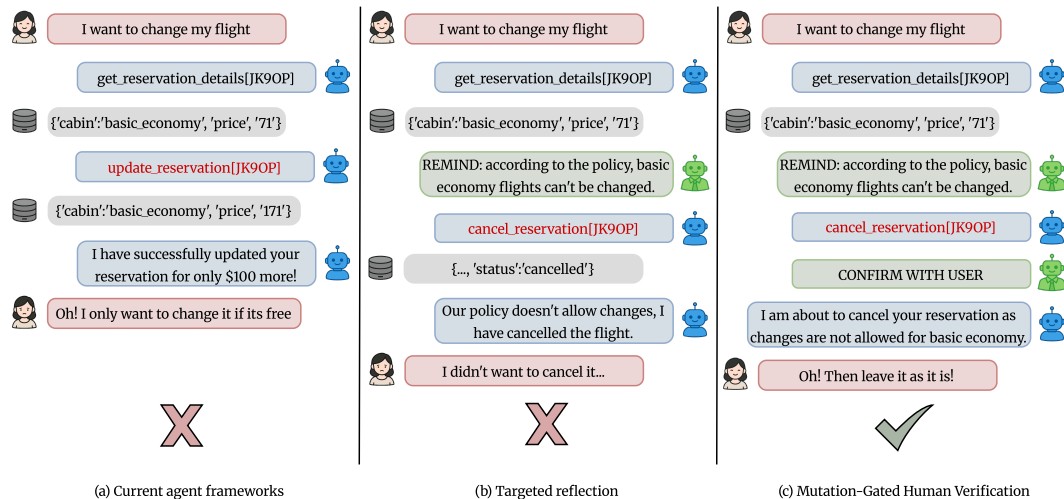

Figure 1: Illustration of Targeted Reflection and Mutation-Gated User Verification

However, efficiency at the mutating gate alone is not enough, because errors grow more frequent as context length increases. Agents progressively lose fidelity to their intended role and policies due to "lost-in-the-middle" effects and the over-trust of stale tokens (Liu et al., 2023; Laban et al., 2025). Even trajectories that begin correctly can drift, leading to misaligned or outdated tool calls at later mutating steps (Kwan et al., 2024; Wang et al., 2024b; Cemri et al., 2025). To sustain reliability in long-horizon settings, we propose two additional mechanisms: lightweight reminders that sharpen constraints before risky steps, and context cleaning that keeps verification-critical history salient. Together, these measures aim to preserve alignment while keeping intervention selective rather than overwhelming.

Yet drawing such conclusions requires trust in the benchmark itself. We found that $\tau$-Bench, though widely adopted, embeds inconsistencies and underspecified instructions that cap attainable performance and blur differences between models. For example, Airline tasks contain contradictory booking policies, and Retail tasks often omit disambiguation (e.g., "pay with the credit card" despite multiple cards on file). To support rigorous evaluation, we re-audit both domains, correcting annotation errors and clarifying policies. We release the result as $\tau$-**Bench Verified**, a cleaned version that preserves task coverage while removing systemic flaws. On this stronger foundation, differences between strong models re-emerge and the benefits of safeguards become clearer, as shown by the consistent gains in Table 6.1.

Building on this foundation, we propose SABER, a model-agnostic, gradient-free safeguard that combines three mechanisms: *Mutation-Gated User Verification* to place decisive steps under direct scrutiny, *Targeted Reflection* to counter "lost-in-the-middle" drift, and *Block-based Context Cleaning* to prevent stale confirmations from crowding and to keep only verification-critical and goal-salient history. We illustrate *Mutation-Gated User Verification* and *Targeted Reflection* in Figure 1, while we detail *Block-based Context Cleaning* in Section 4.

We evaluate SABER across open and closed models, pairing main models with auxiliary models: the main model generates actions, while the auxiliary model provides reflection, verification, and context cleaning. Across $\tau$-Bench Verified and SWE-Bench Verified, SABER consistently yields substantial improvements. For example, `Qwen3-Thinking-235B` paired with a `Qwen3-Instruct-235B` auxiliary improves **19.7%** on Airline Verified, **10.8%** on Retail Verified and **7%** on SWE-Bench Verified, while both `GPT-5` and `Claude 4` gain further headroom once benchmark flaws are removed. These results highlight not only the importance of focusing oversight on mutating steps but also the broader need for refined evaluation to reveal genuine differences in model robustness.

## 2 RELATED WORK

**Agency in AI systems.** While classical AI defined agents broadly as entities that perceive and act on their environment (Russell & Norvig, 1995), recent work views agency as a spectrum of capabilities (Zhang et al., 2024b; Kapoor et al., 2024), emphasizing autonomous goal pursuit, natural language interaction, and structured tool use (OpenAI, 2025). This perspective has driven the development of numerous benchmarks for evaluating LLM capabilities in real-world scenarios (Jimenez et al., 2024; Yao et al., 2024; Wang et al., 2025a; Patil et al., 2025), as well as systems that combine prompt engineering and context engineering techniques to improve agent reliability (AWS, 2024; Liu et al., 2024; AWS / Kiro Team, 2025; Mei et al., 2025; Wang et al., 2025b; Yang et al., 2024; Wang et al., 2024a). In contrast to prior work that focuses on building new systems to enhance performance, we analyze execution trajectories of existing state-of-the-art models to ask a fundamental question: *Do all actions contribute equally to task failure?* Our analysis shows that small flaws in *mutable actions* (Section 3) disproportionately drive failures. Leveraging this insight, we develop SABER, the first system that combines enhanced reflection and selective human-agent collaboration to intervene only when supervision is truly necessary.

**Multi-agent systems.** Another line of research explores using LLMs as central controllers for agents that interact with external environments beyond text-only domains (Deng et al., 2023; Xie et al., 2024). Recent work investigates multi-agent systems where multiple specialized agents interact concurrently (Hong et al., 2024; Li et al., 2023), enabling collaboration for complex tasks. Although promising, such systems are often costly, prone to compounding errors, and have not demonstrated consistent gains on standard benchmarks (Zhang et al., 2025; Cemri et al., 2025). By contrast, our approach is gradient-free and single-agent, focusing on reducing critical mistakes. Specifically, SABER integrates an enhanced reflection mechanism and explicitly identifies irreversible actions that humans can approve before their execution.

**User simulators for enhancing AI systems.** A growing body of work explores LLMs as simulators of human characters, ranging from non-player characters in games to agents embedded in human-like societies or collaborative task settings (Kim et al., 2022; Park et al., 2023; Wu et al., 2023; Chen et al., 2024a; Zhang et al., 2024a; Yao et al., 2024; Barres et al., 2025). These efforts demonstrate that LLMs can emulate realistic human interaction patterns, but they have primarily been used for showcasing simulation rather than improving agent reliability. In our work, user simulators play a different role: they provide a scalable way to approximate the human confirmation step required by SABER. Instead of relying on human evaluators, benchmarks such as $\tau$-Bench offer simulated users that allow us to evaluate how SABER integrates human-in-the-loop feedback. This enables us to systematically test how selective confirmation of irreversible actions reduces decisive errors, while preserving efficiency in real-world scenarios.

**Benchmarks for agent evaluation.** A variety of benchmarks have been proposed to evaluate language agents, yet important limitations remain. Stable ToolBench (Qin et al., 2023) mitigates instability from external APIs through a virtual server, but relies on large models for evaluation, leading to high costs and limited scalability. BFCL (Patil et al., 2025) and HammerBench (Wang et al., 2025a) extend evaluation to multi-turn dialogues, but their trajectories are constructed from predefined content and fail to capture under-specified or evolving real-world user goals. $\tau$-Bench (Yao et al., 2024) and $\tau^2$-Bench (Barres et al., 2025) moves closer to realistic evaluation by embedding agents in domain-specific environments with simulated users. However, as we show in Section 5, annotation errors and under-specified instructions cap achievable performance, weakening its diagnostic reliability. We address this gap by releasing $\tau$-**Bench Verified**, a fully revised version of the Airline and Retail domains that corrects dataset inconsistencies and resolves ambiguities. This benchmark provides a more faithful and trustworthy measure of agent capabilities, enabling robust evaluation of SABER and future systems.

## 3 PROBLEM FORMULATION

We introduce a formal framework to analyze decisive errors in LLM-powered agentic systems, distinguishing between environment-mutating and non-mutating actions within a standard turn-based protocol Hong et al. (2024); Li et al. (2023); Wu et al. (2023); Zhang et al. (2025).

**Background.** Consider an LLM-powered single-agent system $\mathcal{M}$ that operates at discrete time steps. At each step, the agent observes the current state and performs exactly one action. Formally,

$$\mathcal{M} = \langle S, A, P \rangle. \tag{1}$$

Here, $S$ is the set of states, $A$ the action set, and $P(s_{t+1} \mid s_t, a_t)$ the transition law. A trajectory is $\tau = (s_0, a_0, s_1, a_1, \ldots, s_T)$, and failure-indicator $Z(\tau) \in \{0, 1\}$ denotes failure (1) or success (0).

**Decisive deviation (comparative).** Let $\tau^\star$ be a successful reference trajectory for a task ($Z(\tau^\star) = 0$; see Section 5). Let $\tau'$ be a candidate trajectory for the same task, and let $t$ be the earliest index at which their action sequences diverge (prefixes up to $t-1$ match). Denote by $\tilde{a}_t$ the additional action appearing at position $t$ in $\tau'$ (relative to $\tau^\star$). Define the decisive-deviation indicator

$$\Delta_t^+(\tau', \tau^\star) = \begin{cases} 1, & \text{if } Z(\tau^\star) = 0 \text{ and } Z(\tau') = 1, \\ 0, & \text{otherwise.} \end{cases} \tag{2}$$

Thus, $\Delta_t^+ = 1$ captures that introducing the $\tilde{a}_t$ at step at $t$ flips a success into a failure.

**Mutating vs. non-mutating insertions.** Partition the action set into mutating and non-mutating subsets: $A^{\mathrm{mut}} \subseteq A$ (actions that change the external environment or user-visible state) and $A^{\mathrm{non}} = A \backslash A^{\mathrm{mut}}$. Our working hypothesis is that decisive flips arise predominantly from *mutating* insertions:

$$\mathbb{P}(\Delta_t^+ = 1 \mid \tilde{a}_t \in A^{\mathrm{mut}}) \gg \mathbb{P}(\Delta_t^+ = 1 \mid \tilde{a}_t \in A^{\mathrm{non}}) \tag{3}$$

**From local deviations to a dataset-level test.** To connect Eq. 2 and Eq. 3 to corpus-level evidence, we audit trajectories via deviations from the reference plan. Let

$$M(\tau) = \sum_k \mathbf{1}[a_k \in A^{\mathrm{mut}}], \qquad N(\tau) = \sum_k \mathbf{1}[a_k \in A^{\mathrm{non}}],$$

and define absolute deviations

$$d_{\mathrm{mut}}(\tau'; \tau^\star) = \left| M(\tau') - M(\tau^\star) \right|, \qquad d_{\mathrm{non}}(\tau'; \tau^\star) = \left| N(\tau') - N(\tau^\star) \right|.$$

Under Eq. equation 3, success should decrease primarily with $d_{\mathrm{mut}}$ after controlling for $d_{\mathrm{non}}$. For example, overshooting by one extra file deletion (*mutating*) is far more likely to cause failure than adding one redundant search query (*non-mutating*).

| Model | Dataset | Mutating distance | | | Non-mutating distance | | | n |
|-------|---------|------|-----|---|------|-----|---|---|
| | | Coef. | OR | p | Coef. | OR | p | |
| GPT-5 (med) | $\tau$-Bench Retail | -1.06 | 0.35 | $< 0.001$ | -0.01 | 0.99 | 0.781 | 690 |
| GPT-5 (med) | $\tau$-Bench Airline | -2.02 | 0.13 | $< 0.001$ | -0.04 | 0.96 | 0.163 | 297 |
| Qwen3-Thinking | $\tau$-Bench Retail | -0.80 | 0.45 | $< 0.001$ | -0.02 | 0.98 | 0.559 | 345 |
| Qwen3-Thinking | $\tau$-Bench Airline | -2.46 | 0.09 | $< 0.001$ | -0.12 | 0.89 | 0.004 | 297 |
| Claude Sonnet 4.0 | $\tau$-Bench Retail | -2.54 | 0.08 | $< 0.001$ | -0.09 | 0.91 | 0.008 | 690 |
| Claude Sonnet 4.0 | $\tau$-Bench Airline | -3.32 | 0.04 | $< 0.001$ | -0.21 | 0.81 | $< 0.001$ | 297 |

Table 1: Logistic regression of task success on mutating and non-mutating distance across models and datasets. Mutating deviations dominate task failure across models and datasets, while non-mutating deviations have inconsistent or negligible effects. "med" indicates medium reasoning effort.

A logistic regression shows that *mutating deviations* are the primary driver of failure, while *non-mutating deviations* matter far less (Table 1). An odds ratio (OR) below 1 means each extra mismatch reduces success; for instance, in $\tau$-Bench Airline with Claude 4, a mutating mismatch cuts the odds of success by $96\%$ (OR $= 0.04$), compared to only $19\%$ for a non-mutating mismatch (OR $= 0.81$). Across models, mutating deviations consistently yield large, significant penalties, confirming our hypothesis (Eq. 3) that failures stem mainly from errors in mutating steps. Our objective is therefore to reduce decisive deviations, $\Pr[\Delta_t^+ = 1]$, by detecting when actions lie in $A^{\mathrm{mut}}$ and checking them against task constraints (system rules, tool schemas, and user requirements; Fig. 1 (a). Safeguards applied only at these high-risk points minimize overhead while directly targeting the main source of failure. The next section introduces SABER, which operationalizes this strategy.

# 4 SABER: SAFEGUARDING AGAINST MUTATING ACTIONS

As shown by the unguarded failure in Fig. 1(a) and the dataset trends in Table 1, decisive deviations cluster at *mutating* steps. These actions account for only 14–18% of total steps (e.g., Qwen3–Airline: 15.5%, Qwen3–Retail: 18.3%) yet dominate failure risk: a single mutating deviation reduces success odds drastically, whereas non-mutating deviations are nearly harmless (Table 1). To safeguard against this failure mode without overwhelming the user, we introduce SABER, a lightweight, model-agnostic context management system that plugs into existing agent loops without retraining.

## 4.1 SAFEGUARDS FOR MUTATING ACTIONS

**Mutation-gated human verification.** SABER requires explicit user confirmation *only before executing mutating actions*, capping interruptions to roughly one in six turns. Non-mutating actions proceed autonomously. This focused scrutiny concentrates scarce user attention on the steps most likely to flip success into failure, reducing decisive errors while keeping verification burden low; cf. Fig. 1 (c). In practice, this gating also prevents prompt-locking or stalling attacks, since the next post-feedback action is executed directly.

**Targeted reflection.** In long trajectories, mutating actions $a_t \in A^{\mathrm{mut}}$ often produce tool calls that are syntactically valid but semantically inconsistent with system constraints, due to "lost in the middle" effects (Liu et al., 2023; Laban et al., 2025). To counteract this, SABER injects a concise, high-salience summary of key instructions *at the point of mutation*. This reduces miscalibrated tool calls and improves alignment (illustrated in Fig. 1 (b)). When reasoning traces are unsupported, the same summary is appended in a ReAct-style format to preserve guidance.

**Block-based context filtering.** Verification turns can inflate dialogue history and cause *context poisoning* (Breunig, 2025), as shown by the growth of confirmation turns in Fig. 1 (c). SABER therefore partitions trajectories into blocks, summarizes them, and retrieves only the most relevant blocks for the current user query. This keeps the effective context compact and pertinent, mitigating poisoning while preserving the benefits of verification. The retrieval budget $N$ is user-configurable to trade off recall and context-window pressure.

## 4.2 SABER SYSTEM IMPLEMENTATION

To operationalize these safeguards, SABER defines two cooperating models: a *main model*, responsible for generating actions, and an *auxiliary model*, responsible for verification, reflection, and context management. This separation keeps the main policy unchanged while allowing the auxiliary to enforce gates and maintain context cheaply.

For each candidate action $a_t$, the auxiliary model checks whether it is mutating. If so, it reformulates the tool call into a concise natural-language summary with essential preconditions and intended effects, and requests user confirmation. The user's feedback is incorporated into the trajectory ($\tau'$), after which the main model produces its next action—executing the tool call if confirmed or revising if rejected. Non-mutating actions bypass verification entirely. To minimize semantically invalid mutations, the auxiliary model injects a distilled reflection of constraints into a `<think>` block (or a ReAct-style format (Yao et al., 2023) when reasoning traces are unsupported). Finally, to counteract context poisoning, the auxiliary model stores block-level summaries $(s_k, e_k)$ and retrieves the $N$ most relevant blocks by embedding similarity to the latest user query. This block-based filtering retains only verification-critical history, mitigating drift without exceeding context limits. In our implementation, embeddings are cached and summaries are short, so the added latency is marginal relative to typical tool calls.

Taken together, mutation-gated verification, targeted reflection, and block-based filtering deliver a narrow, high-yield intervention surface: most turns remain fully autonomous, while the few high-risk mutating steps receive concise guidance and a single confirmation hop, enabling strong accuracy gains with minimal overhead.

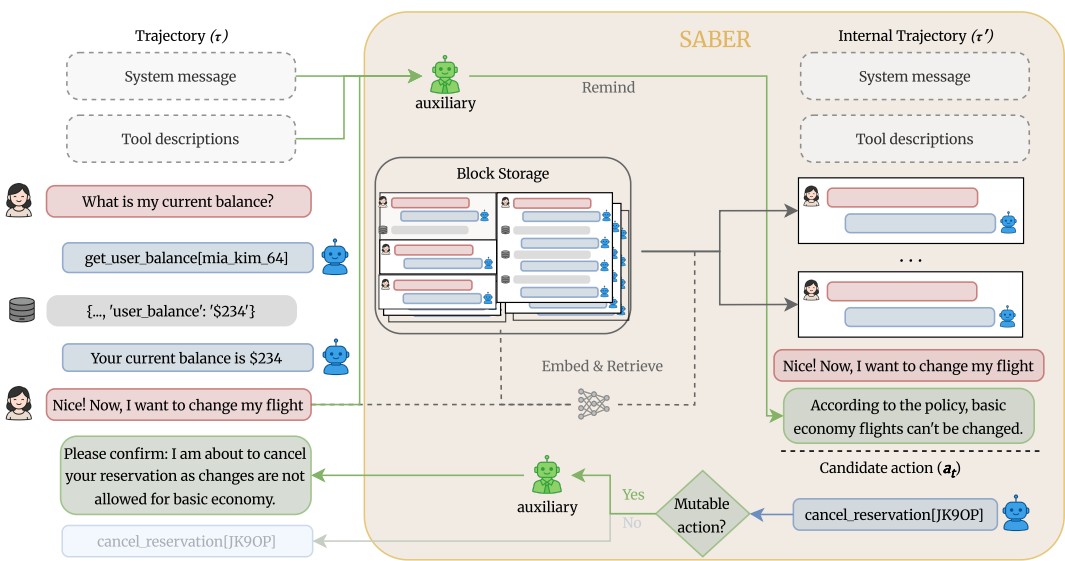

Figure 2: Runtime workflow of SABER. The pipeline is anchored on *mutation-gated human verification*: the auxiliary model inspects whether a candidate action $a_t$ is mutating and, if so, reformulates the tool call into natural language and requests user confirmation. To support this gate in long contexts, the auxiliary model (i) injects a distilled reflection of system instructions and tool constraints into a `<think>` block to reduce invalid mutations, and (ii) applies block-based context filtering to retain only verification-critical and goal-salient history.

## 5  TAU-BENCH VERIFIED

$\tau$-Bench (Yao et al., 2024) is a recently introduced benchmark for evaluating LLM-based agents in realistic interactive environments. It provides domain-specific scenarios (e.g., airline booking, retail shopping) where an agent must complete tasks through multi-turn tool calls while interacting with a simulated user. While valuable for assessing environment interaction, we identified systematic issues in the released datasets that significantly limit achievable performance.

### 5.1  EXISTING ISSUES IN $\tau$-BENCH

In the *Airline* domain, dataset inconsistencies capped performance at roughly **70%**, while in the *Retail* domain, annotation errors limited performance to around **92%**. This is concerning given the widespread use of $\tau$-Bench to evaluate agentic capabilities of state-of-the-art models (Yang et al., 2025; OpenAI, 2025; Anthropic, 2025; MoonshotAI, 2025; Hui et al., 2024). Moreover, these problems persist even in the recently released $\tau^2$-Bench as these domains haven't been updated (Barres et al., 2025). We showcase two representative examples from each domain in Figure 3, a complete set of the ground truth problems can be found in the Appendix C.

### 5.2  UNDER-SPECIFIED USER INSTRUCTIONS.

Beyond ground truth errors, we found that many instructions in the original datasets are underspecified. For instance, in the Airline and Retail domains, 31 out of 50 and 53 out of 115 instructions, respectively, lacked sufficient detail (e.g., a user is asked to pay with "the credit card" despite having two cards on file, but the benchmark accepts only one). Such ambiguities increase benchmark variance and weaken its reliability as a diagnostic tool. While $\tau^2$-Bench acknowledges this issue and introduces a new Telecom domain, it does not resolve the problems in the original Airline or Retail domains.

| (a) $\tau$-Retail example | (b) $\tau$-Airline example |
|---|---|
| **Wiki policy.** Exchanges must involve a *different product option* of the same item. Re-using the exact same option is not allowed. | **Wiki policy.** If a flight is delayed, a certificate can be issued *only after* the reservation is changed or cancelled. |
| **Ground truth (incorrect).** Exchange item ID **8069050545**, with **SAME** item **8069050545** **Error:** Both IDs are identical — violating the rule that exchanges must select a different option. | **Ground truth (incorrect).** 
 • `get_user_details()` 
 • **send_certificate(amount = \$150)** 
 **Error:** A certificate is issued directly, without performing the required change/cancellation. |
| **Correct solution.** Exchange item ID **8069050545**, with different item **3609437808** **Fix:** New product option must differ from the old one. | **Correct solution.** 
 • `get_user_details()` 
 **Fix:** The user doesn't want to change or cancel the flight so no certificate is issued |

Figure 3: Two examples of incorrect ground-truth annotations in $\tau$-Bench. (a) In Retail, the solution reuses the same product ID, violating the policy that exchanges require a different option. (b) In Airline, the solution issues a certificate without first confirming and changing/cancelling the reservation.

## 5.3 $\tau$-BENCH VERIFIED.

To address these shortcomings, we manually reviewed every task, corrected errors, extended user instructions and compiled the revised benchmark, which we term $\tau$-**Bench Verified**. The full list of corrections is included in the Appendix, and we highlight two representative cases in the main text—one from Airline and one from Retail—where annotation issues directly prevented correct model solutions. We release $\tau$-Bench Verified publicly to provide a more faithful benchmark for assessing LLM agent capabilities and to encourage more robust evaluation practices in future work.

## 6 EXPERIMENTS

This section is organized as follows. In Section 6.1, we demonstrate how SABER can enhance the performance of existing models on agentic tasks, such as those in the SWE-bench Verified, Tau-Bench Airline or Tau-Bench Retail benchmark. In Section 5, we conduct a detailed analysis of Tau-Bench Airline and Retail test dataset correctness and propose Tau-Bench Verified.

### 6.1 SABER IMPROVES PERFORMANCE IN AGENTIC TASKS

We observe that every trajectory across benchmarks requires for success one or more mutating actions that can trigger a decisive error (as shown in Eq 2). These mutating actions are necessary for task completion but present an inherent risk. To address this, we apply SABER to improve model performance.

**Models.** SABER is gradient-free and prompt-only, so it applies directly to both closed- and open-weight LLMs. We evaluate `claude-sonnet-4-20250514` and `gpt-5-2025-08-07` (medium reasoning) as closed-source models, and `Qwen3-235B-A22B-Thinking-2507` as an open-weight model. Unless otherwise noted, each model serves as both the *main* agent (actioning) and the *auxiliary* agent (judge/summarizer) within SABER. For Qwen3, we report two auxiliary variants paired with the *Thinking* main: `Qwen3-235B-A22B-Instruct-2507` and

| Benchmark | Qwen3-Thinking-235B | | ChatGPT-5 (med) | | Claude Sonnet 4 | |
|---|---|---|---|---|---|---|
| | No-SABER | SABER | No-SABER | SABER | No-SABER | SABER |
| $\tau$-Bench Airline | 49.3% | **63.3%** | 45.3% | **62.6%** | 51.3% | **56.0%** |
| $\tau$-Bench Retail | 64.3% | **71.6%** | **77.1%** | 76.5% | 73.3% | **78.3%** |
| $\tau$-Bench-V Air | 58.5% | **78.2%** | 78.9% | **82.0%** | 72.1% | **80.3%** |
| $\tau$-Bench-V Ret | 66.9% | **77.7%** | 81.4% | **83.0%** | 82.3% | 81.0% |
| SWE-Bench V | 42.6% | **45.1%** | – | – | – | – |

Table 2: Performance of different models on $\tau$-Bench variants and SWE-Bench Verified, with and without SABER. In the table, "V" stands for verified. To reduce the variance present in $\tau$-Bench, we report the average score over three runs. All evaluations are limited to 30 turns.

`Qwen3-235B-A22B-Thinking-2507`. All models are evaluated on $\tau$-Bench and $\tau$-Bench-Verified (Airline and Retail; see Section 5); additionally, due to the inherent cost of evaluating closed-source models we only evaluate `Qwen3-235B-A22B-Thinking-2507` on SWE-Bench Verified. We use `claude-sonnet-4-20250514` as a simulated user in $\tau$-Bench.

**Baselines and protocol.** We compare each model (and pairing) *with* and *without* SABER. The no-SABER baseline uses each benchmark's standard native tool-calling setup, following prior reports for $\tau$-Bench (Yao et al., 2024; Anthropic, 2025; Yang et al., 2025; OpenAI, 2025). For SWE-Bench Verified, we use `OpenHands` as the tool-calling framework (Wang et al., 2025b; Jimenez et al., 2024). To keep budgets comparable, we cap each episode at 30 turns on every benchmark. We also perform ablations on $\tau$-Bench-Verified (Airline/Retail) using `Qwen3-235B-A22B-Thinking-2507` as the main model to isolate the contribution of each SABER component: (i) remove reflection, (ii) remove mutation-gated verification, and (iii) remove context control. We report both *same-model* pairings (main = auxiliary) and *within-family* mixed pairings to assess sensitivity to the auxiliary model.

We make the following observations:

- **Mutating actions are relatively rare.** Across Airline and Retail, they account for only $\approx 14$–$18\%$ of all steps (e.g., Qwen3–Airline: 15.5%, Claude-4–Retail: 18.1%). This skew means that most of the trajectory proceeds through non-mutating actions, keeping the space of potential interventions small.

- **But when they occur, they carry outsized risk.** A single mutating deviation reduces success odds by 57–82%, while a comparable non-mutating deviation reduces odds by only 7–15% (Table 1). SABER therefore targets verification precisely at these mutating points, minimizing user burden—most steps remain autonomous—while still neutralizing the majority of decisive errors.

- **Large and consistent improvements.** Across both Airline and Retail domains, SABER delivers double-digit gains for the most failure-prone model: Qwen3-Thinking improves by +14.0 pp on Airline (49.3% → 63.3%) and +7.3 pp on Retail (64.3% → 71.6%). Gains also extend to verified settings, with +19.7 pp on $\tau$-Bench-V Air (58.5% → 78.2%) and +10.8 pp on $\tau$-Bench-V Ret (66.9% → 77.7%). More capable baselines still benefit: ChatGPT-5 rises by +17.3 pp on Airline and +3.1 pp on Retail, while Claude Sonnet 4 gains +4.7 pp and +5.0 pp respectively. These consistent lifts across models and datasets (Table 6.1) show that SABER improves weaker and stronger systems alike.

- **Synergy of reflection and verification.** Ablations (Table 6.1) show that reflection and verification each add ∼10 pp in Airline, but together yield the strongest gains (78.7%). This supports our hypothesis: mutating steps require both constraint reminders and user oversight. In Retail, both mechanisms surpass 80% individually, and combined hover around the same score, potentially due to benchmark saturation.

- **Verified benchmarks expose hidden headroom.** Gains are consistently larger on $\tau$-Bench Verified than on the original: for instance, Claude Sonnet 4 jumps +8.2 pp on $\tau$-Bench-V Air versus +4.7 pp on Airline (Table 6.1). Correcting dataset noise thus reveals genuine capacity improvements that standard benchmarks understate.

- **SWE-Bench constraints.** On SWE-Bench Verified, where only the enhanced reflection can be applied, SABER still improves Qwen3-Thinking by about 4 pp, confirming that even partial safeguards matter.

- **Auxiliary–main pairing matters.** The choice of auxiliary model significantly affects outcomes. Using Qwen3-Thinking as the auxiliary yields only a +10 pp gain on $\tau$-Bench-Verified Airline, while pairing the reasoning-focused Qwen3-Thinking main with the instruction-tuned Qwen3-Instruct auxiliary produces much larger improvements (Table 6.1). Systematically exploring which models best complement each other is an open direction that we leave for future work.

| Benchmark | No-SABER | +Reflection | +Verification | Full SABER |
|---|---|---|---|---|
| $\tau$-Bench Verified Airline | 58.0% | 68.0% | 68.7% | 78.7% |
| $\tau$-Bench Verified Retail | 66.9% | 80.8% | 80.5% | 77.7% |

Table 3: Ablation study of SABER on `Qwen3-Thinking-235B` for $\tau$-Bench Verified Airline and Retail. Columns activate different safeguards: *Reflection*, *Mutation-gated user verification*, and their combination in *Full SABER*. In all settings, *Block-based context cleaning* is applied with the number of blocks capped at 16 (see Section 4.2).

The improvements support our formal analysis: decisive deviations are driven primarily by mutating actions (Eq. 3). By gating these actions through simulated user verification and reducing invalid insertions via reflection, SABER lowers the probability that a deviation at step $t$ flips a successful trajectory $\tau^\star$ into a failing one $\tau'$ (Eq. 2), thereby improving overall task success.

# 7 CONCLUSION

This work shows that not all actions are equally risky in long-horizon agent executions: a small slice of *mutating* steps accounts for a disproportionate share of decisive failures. We formalized this with a decisive-deviation test, validated the *mutating-dominates* hypothesis at the dataset level, and re-audited $\tau$-Bench to produce $\tau$-**Bench Verified**, a cleaner yardstick that exposes genuine headroom.

Building on these findings, we introduced SABER, a model-agnostic, gradient-free safeguard that focuses intervention where it pays off most: *mutation-gated user verification* at risky steps, *targeted reflection* to keep tool calls semantically consistent with constraints, and *block-based context cleaning* to keep verification-critical history salient.

Empirically, SABER delivers consistent gains across models and domains on $\tau$-Bench and $\tau$-Bench Verified (Table 6.1). SABER demonstrates that not all actions need equal scrutiny: focusing safeguards on rare but decisive mutating steps yields outsized reliability improvements; gating *only* these steps concentrates user attention while leaving most turns autonomous.

# 8 LIMITATIONS

SABER is introduced as an online safeguard rather than a training-time property. While mutation-gated verification and targeted reflection reduce decisive errors at test time, they are externally imposed. Future training regimes could internalize these behaviors—for example, by shaping loss functions around decisive deviations or teaching models to self-identify mutating actions—so that models regulate risky steps without auxiliary intervention. This would reduce reliance on auxiliary mechanisms and make the safeguards part of the model's native reasoning process.

A second limitation is that SABER depends on access to a user or user simulator for confirming irreversible actions. This assumption matches real-world deployments, but few benchmarks natively support user-in-the-loop verification. Simulated users help approximate the setting, yet they inevitably simplify the variability of human feedback. Expanding benchmark design to include confirmation and reflection episodes is therefore essential for evaluating and advancing practical safeguards, and for ensuring that improvements transfer reliably to real users.

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

## A  BROADER IMPACTS

This work advances the reliability of LLM-powered agents by identifying and mitigating errors concentrated at mutating actions. By introducing lightweight safeguards that require human confirmation only at high-risk points, our approach reduces failure rates without imposing excessive user burden. This makes agentic systems safer and more predictable for deployment in real-world domains such as software engineering, operations, and customer support. More broadly, selective oversight helps prevent costly or harmful unintended changes, thereby supporting responsible adoption of LLM agents while keeping human involvement efficient and focused.

## B  REPRODUCIBILITY STATEMENT.

We release an anonymous repository with code, data, and instructions to fully reproduce our results: https://anonymous.4open.science/r/SABER-1E54/. The repository contains (i) the SABER implementation and configuration used in all experiments (Section 4); (ii) the corrected $\tau$-Bench Verified datasets (Airline/Retail), along with scripts and diffs documenting each fix (Section 5, Appendix A); (iii) experiment drivers, prompts, and evaluation pipelines for all models and settings reported (Section 6.1); (iv) scripts to regenerate figures and tables, including the logistic-regression analyses in Table 1.

## C  APPENDIX: CORRECTIONS TO $\tau$-BENCH–AIRLINE

We audited the Airline split and found several tasks whose *action traces* (or their parameters / allowed paths) violated the official Wiki policy. For each task below, we (i) restate the governing rule, (ii) show the *previous* (incorrect) ground truth action(s) from the original dataset, (iii) show the *correct* action(s) after our fix (only actions are shown; instruction-only edits are excluded), and (iv) explain the error and the fix in detail.

**Legend.** We reference the Airline Wiki sections: *Book flight*, *Modify flight*, *Cancel flight*, *Refund*, baggage allowances, and single-call constraints (one tool call *or* a user reply at a time).

TASK 1 — PAYMENT METHOD ALIGNMENT (GIFT CARD CHOICE)

> **Wiki policy.** Payment ordering and limits: at most one certificate, one credit card, and up to three gift cards; use user-specified method/order when feasible.

> **Ground truth (incorrect).**
> ```
> update_reservation_baggages(OBUT9V, total_baggages=2, nonfree_baggages=0,
> payment_id=gift_card_6276644)
> ```

> **Correct solution.**
> ```
> update_reservation_baggages(OBUT9V, total_baggages=2, nonfree_baggages=0,
> payment_id=gift_card_7480005)
> ```

**Why it is wrong.**  The chosen gift card ID did not reflect the user's stated order (use the smallest-balance card first), violating payment preferences under the policy.

**Fix explanation.**  Swap to the correct gift card consistent with the instruction and payment constraints; keep add-only baggage semantics unchanged.

TASK 2 — ENCODE MULTIPLE COMPLIANT SEQUENCES (VALID_ACTION_PATHS)

> **Wiki policy.** Flight & passenger edits may be done in different valid orders; annotate only
> *policy-compliant* sequences. Baggage is add-only; keep one payment method per modification.

**Ground truth (incorrect).**
A single hard-coded sequence:
```
update_reservation_flights(FQ8APE, cabin=economy,
flights=[HAT056@2024-05-25, HAT138@2024-05-25],
payment=gift_card_8190333)
update_reservation_passengers(FQ8APE, passengers=[Omar Rossi
(1970-06-06)])
update_reservation_baggages(FQ8APE, total_baggages=3, nonfree_baggages=0,
payment=gift_card_8190333)
```

**Correct solution.**
```
valid_action_paths={ three policy-equivalent orders over update_reservation_flights,
update_reservation_passengers, update_reservation_baggages (all with the same
arguments as above) }; actions=[].
```

**Why it is wrong.** Only one sequence was accepted even though multiple policy-compliant orders exist; this unfairly penalizes correct agent behavior that follows a different but valid order.

**Fix explanation.** Replace the single sequence with `valid_action_paths` enumerating all compliant permutations that (i) upgrade to economy if needed, (ii) set passenger to self, (iii) add 3 bags, and (iv) use the specified gift card.

TASK 3 — FLIGHT NUMBERS/DAY-AFTER RULE

**Wiki policy.** On modifications, keep endpoints and trip type; honor date constraints precisely; if the user requests "same numbers next day", preserve the numbers and shift dates only.

**Ground truth (incorrect).**
Drifted from the "day-after" constraint and/or altered flight numbers in the action payload.

**Correct solution.**
```
update_reservation_flights(M05KNL, cabin=economy,
    flights=[HAT227@2024-05-24, HAT139@2024-05-24],
    payment=gift_card_8887175)
```

**Why it is wrong.** Any deviation in flight numbers or an incorrect date shift breaks the user's explicit "next-day" directive.

**Fix explanation.** Lock numbers; move them exactly one day later; keep cabin unchanged and a single refund/payment method per policy.

TASK 4 — CANCEL+REBOOK PATH AND BAGGAGE ENUMERATION

**Wiki policy.** If basic economy cannot be modified or payment constraints prevent combining methods, use *cancel + book* with explicit payment ordering (certificates → gift cards → credit card). Baggage is add-only.

**Ground truth (incorrect).**
A single linear path:
```
cancel_reservation(K1NW8N)
book_reservation(JFK→SFO, round_trip, cabin=business,
    flights=[HAT023@05-26, HAT204@05-28, HAT100@05-28],
    passengers=[Mohamed, Raj, Liam],
    payment_methods=[certificate_3765853:500, gc_8020792:198,
gc_6136092:129, cc_2198526:1786],
    total_baggages=0, nonfree_baggages=0, insurance=no)
```

**Correct solution.**

```
valid_action_paths={ all begin with cancel_reservation(K1NW8N) then
book_reservation(...) with enumerated total_baggages in {0,3,6,9}, identical
flights/passengers/payments as above, insurance=no }; actions=[].
```

**Why it is wrong.**   Only one baggage outcome was hard-coded. The Wiki allows add-only baggage and real agents may result in different (still valid) totals. Penalizing those paths is incorrect.

**Fix explanation.**   Encode the shared cancel+book logic but enumerate the allowed baggage totals via `valid_action_paths`, preserving payment order and limits.

TASK 5 — PASSENGER MAPPING FOR MULTI-CERTIFICATE WORKAROUND

**Wiki policy.** One certificate per reservation; to use multiple certificates, split into multiple bookings with correct passenger identities.

**Ground truth (incorrect).**
After `cancel_reservation(K1NW8N)` three bookings mis-assign passengers:
```
book_reservation(..., passengers=[Aarav Sanchez])
book_reservation(..., passengers=[Evelyn Wilson])
```

**Correct solution.**
```
book_reservation(..., passengers=[Raj Sanchez])
book_reservation(..., passengers=[Liam Wilson])
```
(Other fields unchanged; 0 checked bags enforced.)

**Why it is wrong.**   The split-reservation trick must preserve correct passenger identities per instruction; mislabeling breaks both user intent and safety checks.

**Fix explanation.**   Rename affected passengers to `Raj` and `Liam` to match the instruction while keeping single-certificate-per-PNR semantics.

TASK 6 — AVOID PREMATURE CANCELLATION

**Wiki policy.** Only cancel after confirming eligibility (within 24h, insurance, etc.) and after the user agrees; avoid auto-cancel when a modification might suffice.

**Ground truth (incorrect).**
`cancel_reservation(H9ZU1C)` *followed by* `book_reservation(...)`

**Correct solution.**
(No auto-cancel.) Keep only the booking flow once the user confirms the search outcome; cancellation remains conditional and user-driven.

**Why it is wrong.**   Auto-cancel violates the Wiki's cancellation prerequisites and removes the user's chance to accept a modification-first solution.

**Fix explanation.**   Remove auto-cancel from the gold trace; agents must first attempt removal or present options, then cancel only with eligibility and explicit confirmation.

TASK 7 — ENCODE ACCEPTABLE OUTCOMES (BAGS 0 VS 3)

**Wiki policy.** For new bookings, passenger list must match intent; baggage add-only; payment limits must be respected. Accept multiple valid bag totals if both comply.

**Ground truth (incorrect).**

```
Single outcome: book_reservation(..., passengers=[Ivan Smith],
payments=[gc_8516878:128, cc_3563913:247], total_baggages=0,
nonfree_baggages=0, insurance=no)
```

**Correct solution.**
`valid_action_paths`={ same booking with either `total_baggages=0` *or* `total_baggages=3` (nonfree=0), payments unchanged }; `actions=[]`.

**Why it is wrong.** The dataset disallowed another equally valid end-state (3 free bags), unfairly penalizing compliant agents.

**Fix explanation.** Adopt `valid_action_paths` to encode both allowed baggage totals without changing passenger/payment correctness.

TASK 8 — REMOVE UNRELATED SEARCHES/CALCULATIONS

**Wiki policy.** Tool calls must be necessary and policy-aligned; baggage is add-only and may leverage tier benefits even if upgrades fail.

**Ground truth (incorrect).**
`get_reservation_details(YAX4DR); search_direct_flight(BOS→MCO);`
`search_direct_flight(MCO→MSP); calculate(...);` *then*
`update_reservation_baggages(...)`

**Correct solution.**
`update_reservation_baggages(YAX4DR, total_baggages=2, nonfree_baggages=0,`
`payment_id=...)`

**Why it is wrong.** The search/calculation calls are unrelated to the asked change and introduce spurious side-effects.

**Fix explanation.** Retain only the necessary baggage update leveraging Gold benefits; drop unrelated tool calls.

TASK 9 — VERIFY THEN CANCEL (EXPLICIT PNR FLOW)

**Wiki policy.** On cancellation, first fetch reservation details (PNR), then proceed with refund-to-original-payment if eligible.

**Ground truth (incorrect).**
`actions=[]` (no explicit verification/cancellation calls)

**Correct solution.**
`get_reservation_details(GV1N64); cancel_reservation(GV1N64)`

**Why it is wrong.** The gold trace lacked the required *verify-then-cancel* sequence, obscuring the correct policy flow.

**Fix explanation.** Add explicit detail retrieval before cancellation and ensure refund routing to the original payment method.

TASK 10 — NO PROACTIVE COMPENSATION ON DELAYS

**Wiki policy.** Delay compensation is a *gesture* only after confirming facts *and* when the user *changes or cancels* the reservation. Do not issue certificates otherwise.

**Ground truth (incorrect).**
`send_certificate(user=ethan_martin_2396, amount=$150)`

**Correct solution.**
```
get_user_details(ethan_martin_2396);get_reservation_details(M61CQM)
```

**Why it is wrong.** A certificate was issued without the required change/cancel step.

**Fix explanation.** Remove compensation; only confirm user and PNR. (The user did not want to change/cancel.)

TASK 11 — CHARGE THE SPECIFIED CARD ON DATE PUSH

**Wiki policy.** For changes, collect payment/refund method; reflect user-specified payment instrument.

**Ground truth (incorrect).**
```
actions=[]
```

**Correct solution.**
```
update_reservation_flights(4NQLHD, cabin=economy,
   flights=[HAT190@05-24, HAT047@05-24, HAT021@05-26, HAT279@05-27],
   payment_id=credit_card_7434610)
```

**Why it is wrong.** The gold trace omitted the concrete change and the requested card.

**Fix explanation.** Include the multi-segment date push and the specified card for the fare difference, subject to user's ¡1000$confirmation$.

TASK 12 — CANCEL+REBOOK FOR AIRPORT CHANGE (PLUS FREE BAG)

**Wiki policy.** Changing airports is often safer as cancel+rebook; baggage charging must respect allowances.

**Ground truth (incorrect).**
```
update_reservation_flights(VA5SGQ, cabin=economy, flights=[HAT169@05-17,
HAT033@05-19], payment=cc_8003957)
update_reservation_baggages(VA5SGQ, total_baggages=1, nonfree_baggages=1,
payment=cc_8003957)
```

**Correct solution.**
```
get_reservation_details(VA5SGQ);cancel_reservation(VA5SGQ);
book_reservation(user=raj_brown_5782, DTW↔JFK, round_trip,
cabin=economy,
   flights=[HAT169@05-17, HAT033@05-19], passengers=[Raj Brown],
   payment_methods=[cc_8003957:311], total_baggages=1, nonfree_baggages=0,
insurance=no)
```

**Why it is wrong.** Applying a JFK switch as a direct *modify* is brittle; also, charging a non-free bag contradicts the allowance in this context.

**Fix explanation.** Move to cancel+book and ensure the bag is free (`nonfree_baggages=0`); keep explicit payment and PNR verification.

TASK 13 — CAPPED-CHANGE FALLBACK TO NEW BE BOOKING

**Wiki policy.** If change fee exceeds the user cap, book a new *basic economy* ticket using the specified payment order; no insurance or bags added post hoc.

**Ground truth (incorrect).**
```
actions=[]
```

> **Correct solution.**
> ```
> book_reservation(user=daiki_lee_6144, JFK→DTW, one_way,
> cabin=basic_economy,
>    flights=[HAT092@2024-05-17], passengers=[Daiki Lee (1976-10-08)],
>    payment_methods=[gc_3112961:51, cc_6198952:3], total_baggages=0,
> nonfree_baggages=0, insurance=no)
> ```

**Why it is wrong.** The gold trace lacked the required fallback path that respects the user's $100 cap.

**Fix explanation.** Emit the explicit BE booking with the correct payment ordering and zero add-ons.

TASK 14 — ENCODE TWO COMPLIANT NONSTOP-CHANGE PATHS

> **Wiki policy.** For BE tickets, upgrade to economy before changing; reflect user fee cap; resist unnecessary human transfer.

> **Ground truth (incorrect).**
> A single fixed flow mixing search + update sequences, without recognizing alternative valid orders.

> **Correct solution.**
> ```
> valid_action_paths={ two sequences beginning with
> get_user_details(ivan_rossi_8555);get_reservation_details(OWZ4XL);
> search_direct_flight(EWR→LAX@05-21);
> ```
> then either `update_reservation_flights(...  flights=[HAT202,HAT232],`
> `payment=cc_9659780)` *then* `update_reservation_flights(...  flights=[HAT041],`
> `payment=cc_9659780)`
> *or* directly `update_reservation_flights(...  flights=[HAT041],`
> `payment=cc_9659780) }`.

**Why it is wrong.** Only one rigid path was accepted, penalizing agents that chose another compliant order.

**Fix explanation.** Capture both valid orders as `valid_action_paths` with the same policy-compliant parameters.

TASK 15 — FREE VS NON-FREE BAG CORRECTION

> **Wiki policy.** Baggage charges must reflect cabin/tier allowances; do not charge when eligible for free bags.

> **Ground truth (incorrect).**
> ```
> update_reservation_baggages(HXDUBJ, total_baggages=2, nonfree_baggages=2,
> payment=gift_card_6941833)
> ```

> **Correct solution.**
> ```
> update_reservation_baggages(HXDUBJ, total_baggages=2, nonfree_baggages=0,
> payment=gift_card_6941833)
> ```

**Why it is wrong.** It billed for bags that should have been free under the chosen path.

**Fix explanation.** Set `nonfree_baggages=0`, preserving the rest of the payload.

TASK 16 — OFFER MULTIPLE SECOND-CHEAPEST OPTIONS (VALID_ACTION_PATHS)

> **Wiki policy.** When the user requests "second-cheapest", it may admit multiple equivalent flight pairs; encode compliant options; keep card-only payment.

**Ground truth (incorrect).**
Single path: `book_reservation(JFK→SFO, one_way, cabin=economy, flights=[HAT235,HAT268], passengers=[Aarav Ahmed(1981-05-26)], payment_methods=[cc_9074831:260], total_baggages=2, nonfree_baggages=0, insurance=no)`

**Correct solution.**
`valid_action_paths=`{ two `book_reservation` variants (the above and another with flights=[HAT069,HAT258], same passenger/DOB/card]; different baggage totals {0,2} allowed) };
`actions=[]`.

**Why it is wrong.** Only one acceptable second-cheapest option was encoded.

**Fix explanation.** Permit multiple policy-identical choices via `valid_action_paths`, keeping card-only payment and other constraints.

TASK 17 — REMOVE PREMATURE CANCELLATION (MULTI-RESERVATION EDIT)

**Wiki policy.** Do not cancel before confirming eligibility; when editing multiple PNRs, fetch details and then apply modifications with explicit payment flow.

**Ground truth (incorrect).**
`cancel_reservation(NQNU5R)` *present by default* alongside other search/update calls.

**Correct solution.**
`get_reservation_details(M20IZO); search_direct_flight(...); update_reservation_flights(...)` (*no* auto-cancel)

**Why it is wrong.** The trace cancelled without verifying the window/insurance or confirming with the user.

**Fix explanation.** Drop the eager cancel call; retain targeted search/update only.

TASK 18 — SAME (REMOVE EAGER CANCEL & STRAY ARITHMETIC)

**Wiki policy.** Keep action traces minimal and necessary; no stray `calculate` calls in gold traces unless required by the API.

**Ground truth (incorrect).**
`cancel_reservation(NQNU5R)` and `calculate(430+412-(136+109))`

**Correct solution.**
Only the relevant
`get_reservation_details/search_direct_flight/update_reservation_flights` calls.

**Why it is wrong.** Spurious cancellation and arithmetic are not part of the required modify flow.

**Fix explanation.** Remove both; keep only policy-relevant actions.

TASK 19 — DON'T PICK A RESERVATION TO CANCEL IN ADVANCE

**Wiki policy.** When duplicate-day flights exist, the agent must confirm which one to cancel; no preselected PNR.

**Ground truth (incorrect).**
`cancel_reservation(9HBUV8)` present by default.

**Correct solution.**
Only the detail lookups: `get_user_details`, `get_reservation_details(...)` (no auto-cancel).

**Why it is wrong.** It assumes the target PNR without agent confirmation.

**Fix explanation.** Remove the cancellation; let the conversation identify the correct PNR first.

TASK 20 — ONE FREE CHECKED BAG HONORED

**Wiki policy.** Respect free-bag allowance; do not charge when a bag is free.

**Ground truth (incorrect).**
`book_reservation(..., payment_methods=[certificate_8045380:348],`
`total_baggages=0, nonfree_baggages=0)`

**Correct solution.**
`book_reservation(..., payment_methods=[certificate_8045380:348],`
`total_baggages=1, nonfree_baggages=0)`

**Why it is wrong.** The user explicitly wanted one *free* checked bag (allowance permits it).

**Fix explanation.** Set `total_baggages=1` and keep `nonfree_baggages=0`.

TASK 21 — REQUIRE DURATIONS BEFORE CANCEL/UPGRADE

**Wiki policy.** The agent must present flight durations (including layovers) before cancellation/upgrade choices; do not cancel/upgrade preemptively.

**Ground truth (incorrect).**
Contains `cancel_reservation(S61CZX)` and an extra `update_reservation_flights(...)` for KC18K6 before the user decides.

**Correct solution.**
Keep only the detail and search calls necessary to present options; defer any `cancel_reservation`/`update_reservation_flights` until after the user chooses (per durations).

**Why it is wrong.** It executes irreversible actions before the user can review durations, violating the decision loop.

**Fix explanation.** Strip premature actions; require an explicit user pick *after* durations are shown.

TASK 22 — UPGRADE-TO-BUSINESS-FIRST RULE (THEN CANCEL)

**Wiki policy.** If a reservation has BE segments and the user insists on canceling but policy requires upgrade-first, upgrade cabin to *business* before cancellation (with specified card).

**Ground truth (incorrect).**
`update_reservation_flights(XEHM4B, cabin=economy, ...)`; stray
`calculate(...)` calls.

**Correct solution.**
`update_reservation_flights(XEHM4B, cabin=business,`
`flights=[HAT005@05-20, HAT178@05-30])`
(Use CC ending 2135 for cabin-difference per policy; remove `calculate` calls.)

**Why it is wrong.** It applied the wrong cabin for the upgrade-first requirement and included irrelevant arithmetic calls.

**Fix explanation.** Upgrade to *business* before cancellation, charge the stated card, and remove non-essential calculations.

TASK 23 — NO COMPENSATION WHILE KEEPING THE FLIGHT

> **Wiki policy.** For delay complaints without a change/cancel, do not issue a certificate; confirm facts only.

> **Ground truth (incorrect).**
> `get_user_details(noah_muller_9847);get_reservation_details(SDZQKO);`
> `get_reservation_details(4OG6T3);send_certificate($50)`

> **Correct solution.**
> `actions=[]` (the conversation tests the agent's ability to confirm facts; no compensation nor changes requested)

**Why it is wrong.** It offers compensation where policy forbids it.

**Fix explanation.** Remove certificate issuance; keep to fact confirmation in dialogue.

TASK 24 — APPLY THE PHONE APPROVAL (VERIFY THEN CANCEL)

> **Wiki policy.** If the user asserts prior phone approval, still verify PNR and then attempt cancellation; avoid human transfer unless strictly required.

> **Ground truth (incorrect).**
> `get_user_details(raj_sanchez_7340);get_reservation_details(MZDDS4)` (*no cancellation step*)

> **Correct solution.**
> `get_user_details(raj_sanchez_7340);get_reservation_details(MZDDS4);`
> `cancel_reservation(MZDDS4)`

**Why it is wrong.** The trace never attempted the cancellation even after verification.

**Fix explanation.** Add the explicit cancel call to reflect the user's prior approval and the Wiki's scoped authority.

# D APPENDIX: CORRECTIONS TO $\tau$-BENCH—RETAIL (ACTION-LEVEL ANNOTATIONS)

We audited the Retail split and found several tasks whose *action traces* (or allowed paths) violated the Wiki policy. For each task below, we (i) restate the governing rule, (ii) show the *previous* (incorrect) ground-truth actions from the original dataset, (iii) show the *current* (correct) actions after our fix (only actions are shown; instruction-only edits are excluded), and (iv) explain the error and the fix in detail.

**Legend.** We reference Retail Wiki sections: order lookup, address edits (user vs order), exchange/return/cancel tools, refund/payment routing, and `valid_action_paths` when multiple policy-compliant sequences exist. One tool call *or* a user reply per step.

TASK 1 — REMOVE PREMATURE RETURN BEFORE TRANSFER

> **Wiki policy.** Do not execute irreversible actions (returns/cancellations) once the user escalates to a human; transfer should end the bot's action flow.

> **Ground truth (incorrect).**

```
...    get_user_details; get_order_details #W5490111; get_order_details
#W7387996;
return_delivered_order_items(#W5490111, item_ids=[4579334072, 6117189161,
4947717507], payment_method_id=paypal_9497703)
transfer_to_human_agents(reason=refund_not_paypal, priority=high)
```

**Correct solution.**
```
...    get_user_details; get_order_details #W5490111; get_order_details
#W7387996;
transfer_to_human_agents(reason=refund_not_paypal, priority=high)
```

**Why it is wrong.**  A full return was executed *before* escalation, contradicting the handoff boundary.

**Fix explanation.**  Remove the return call; once the conversation escalates, no further irreversible actions should occur in the gold trace.

TASK 2 — SAME AS TASK 1 IN ALTERNATE TEMPLATE

**Wiki policy.** Same as Task 1; no irreversible actions after handoff.

**Ground truth (incorrect).**
```
...    get_user_details; get_order_details #W5490111; get_order_details
#W7387996;
return_delivered_order_items(#W5490111, item_ids=[4579334072, 6117189161,
4947717507], payment_method_id=paypal_9497703)
...
```

**Correct solution.**
```
...    get_user_details; get_order_details #W5490111; get_order_details
#W7387996;
(no early return call kept)
```

**Why it is wrong.**  Same violation as Task 1.

**Fix explanation.**  Drop the premature return so the escalation path is clean.

TASK 3 — EXCHANGE TO *cheapest* VERSION (CORRECT TARGET)

**Wiki policy.** When user requests the cheapest replacement, target the correct SKU; do not 'exchange' to the same item.

**Ground truth (incorrect).**
```
exchange_delivered_order_items(#W2890441, item_ids=[8069050545],
new_item_ids=[8069050545], payment_method_id=credit_card_1061405)
```

**Correct solution.**
```
exchange_delivered_order_items(#W2890441, item_ids=[8069050545],
new_item_ids=[3609437808], payment_method_id=credit_card_1061405)
```

**Why it is wrong.**  The "exchange" kept the same SKU, failing to honor "cheapest available" constraint.

**Fix explanation.**  Point the exchange to the cheapest valid SKU (`3609437808`).

TASK 4 — USE THE CORRECT `GET_ITEM_DETAILS` TOOL

**Wiki policy.** Use item-level detail tool for cart SKUs; product-level tool is for catalog browsing.

**Ground truth (incorrect).**
```
get_product_details(4107812777); get_product_details(1421289881);
get_product_details(4107812777)
```

**Correct solution.**
```
get_item_details(4107812777); get_item_details(1421289881);
get_item_details(4107812777)
```

**Why it is wrong.** The wrong introspection tool risks mismatched pricing/specs versus the order line items.

**Fix explanation.** Swap to `get_item_details` for order-linked SKUs.

TASK 5 — ADDRESS EDITS REQUIRE ORDER CONFIRMATION PATHS

**Wiki policy.** When both user default and pending order addresses change, encode *valid sequences*; revert only after confirming order-level updates.

**Ground truth (incorrect).**
```
find_user_id_by_name_zip(...);modify_user_address(to=NY);get_order_details
#W4967593, #W9911714, #W5733668;
modify_pending_order_address(#W9911714, to=NY);
modify_user_address(back=CO)
```

**Correct solution.**
```
valid_action_paths=[
  [find_user..., modify_user_address(to=NY),
modify_pending_order_address(#W9911714,to=NY),
modify_user_address(back=CO)],
  [find_user..., modify_user_address(to=NY),
modify_user_address(back=CO),
modify_pending_order_address(#W9911714,to=NY)]
]; actions=[]
```

**Why it is wrong.** A single rigid order of operations over-constrains compliant agent behavior.

**Fix explanation.** Enumerate both acceptable sequences via `valid_action_paths`; clear the strict `actions` list.

TASK 6 — "CANCEL ONLY THE HOSE" + RETURNS ACROSS ORDERS

**Wiki policy.** Encode the user's constraints precisely (cancel one pending item only; return specified delivered items); allow equivalent call orders via `valid_action_paths`.

**Ground truth (incorrect).**
```
actions=[find_user_id_by_name_zip; get_user_details; get_order_details
#W3792453,#W7181492,#W5565470,#W2575533;
return_delivered_order_items(...  #W3792453:  [4293355847]);
return_delivered_order_items(...  #W7181492:  [5753502325,9851293632]);
return_delivered_order_items(...  #W5565470:  [9570044148,6857426243]);
get_order_details #W2575533; calculate("200.8 + 96.35 + 193.38 + 231.37
+ 196.53")]
```

**Correct solution.**
```
valid_action_paths=[ five permutations of the three return calls above,
each preceded by lookups and followed by the same calculate call;
(the pending hose is checked but not cancelled if it would require
whole-order cancel)
]; actions=[]; outputs=["918.43"]
```

**Why it is wrong.** Only one rigid sequence was allowed and the "cancel just the hose" constraint wasn't structurally enforced.

**Fix explanation.** Use `valid_action_paths` to accept any policy-equivalent ordering of the returns while preserving the single-item cancel restriction.

TASK 7 — TWO-WAY EXCHANGE PLAN (BAMBOO SKATEBOARD + HOSE SKU)

> **Wiki policy.** When multiple exchanges must both occur, encode either order as valid; target the exact SKUs.

> **Ground truth (incorrect).**
> ```
> actions=[find_user; get_user_details; get_order_details #W3792453; ...;
> return_delivered_order_items(...);...] (returns instead of exchanges)
> ```

> **Correct solution.**
> ```
> valid_action_paths=[
>   [find_user; get_user_details; get_order_details #W3792453;
> get_product_details(1968349452);
>   exchange_delivered_order_items(#W3792453, [4293355847]→[8176740019]);
>   exchange_delivered_order_items(#W7181492, [5753502325]→[5206946487])
> ],
>   [find_user; get_user_details; ...; exchange #W7181492; exchange
> #W3792453 ]
> ]; actions=[]; outputs=["180.1","189.57","208.6"]
> ```

**Why it is wrong.** Returns were used instead of the requested SKU-for-SKU exchanges; only one ordering was accepted.

**Fix explanation.** Replace with exchanges and allow both execution orders via `valid_action_paths`.

TASK 8 — REMOVE STRAY ARITHMETIC IN CANCEL FLOW

> **Wiki policy.** Gold traces must avoid unrelated `calculate` calls; show only necessary cancellation steps.

> **Ground truth (incorrect).**
> ```
> ...  get_order_details #W2702727; get_order_details #W8268610;
> calculate("164.28"); cancel_pending_order(#W8268610,
> reason=no_longer_needed)
> ```

> **Correct solution.**
> ```
> ...  get_order_details #W2702727; get_order_details #W8268610;
> cancel_pending_order(#W8268610, reason=no_longer_needed)
> ```

**Why it is wrong.** The arithmetic call is extraneous and not required by the API.

**Fix explanation.** Drop `calculate`; keep minimal lookup → cancel sequence.

TASK 9 — ENCODE ALTERNATIVE CANCEL REASONS

> **Wiki policy.** When multiple policy-acceptable reasons exist, allow them via `valid_action_paths`.

> **Ground truth (incorrect).**
> ```
> actions=[find_user; get_user_details; get_order_details #W2417020;
> cancel_pending_order(#W2417020, reason=no_longer_needed)]
> ```

```
Correct solution.
valid_action_paths=[
  [find_user; get_user_details; get_order_details #W2417020;
cancel_pending_order(reason=ordered_by_mistake)],
  [find_user; get_user_details; get_order_details #W2417020;
cancel_pending_order(reason=no_longer_needed)]
]; actions=[]
```

**Why it is wrong.**  Only one acceptable reason was encoded.

**Fix explanation.**  Permit either reason through `valid_action_paths`.

TASK 10 — ORDER CANCEL AND LAPTOP EXCHANGE: TWO VALID ORDERS

**Wiki policy.** Multiple compliant sequences (cancel first vs exchange first) must be accepted when independent.

```
Ground truth (incorrect).
actions=[cancel_pending_order(#W3189752);
modify_pending_order_items(#W5166363, [3334537816]→[3265035808],
payment=credit_card_4466831)]
```

```
Correct solution.
valid_action_paths=[
  [cancel_pending_order(#W3189752); modify_pending_order_items(#W5166363,
[3334537816]→[3265035808], cc_4466831)],
  [modify_pending_order_items(#W5166363, ...);
cancel_pending_order(#W3189752)]
]; actions=[]
```

**Why it is wrong.**  Only one action order was accepted, penalizing otherwise-correct paths.

**Fix explanation.**  Encode both compliant orders in `valid_action_paths`.

TASK 11 — WATER BOTTLE: CORRECT REPLACEMENT SKU

**Wiki policy.** Exchanges must target the precise requested variant (capacity/color/material constraints).

```
Ground truth (incorrect).
modify_pending_order_items(#W8661412, item_ids=[3453331371],
new_item_ids=[2439754078], payment=credit_card_7239357)
```

```
Correct solution.
modify_pending_order_items(#W8661412, item_ids=[3453331371],
new_item_ids=[7661609223], payment=credit_card_7239357)
```

**Why it is wrong.**  The target SKU didn't match the intended 1000ml variant (with color constraint).

**Fix explanation.**  Point to the correct 1000ml SKU `7661609223`.

TASK 12 — CANCEL "ANY ORDER CONTAINING X" (TWO ORDERS)

**Wiki policy.** When the user allows canceling any order that includes certain items, either order of per-order cancellations is fine.

```
Ground truth (incorrect).
actions=[cancel_pending_order(#W3289292);
cancel_pending_order(#W9722559)]
```



**Correct solution.**
```
valid_action_paths=[ [cancel(#W3289292); cancel(#W9722559)],
[cancel(#W9722559); cancel(#W3289292)] ]; actions=[]
```


**Why it is wrong.** One rigid order only.

**Fix explanation.** Allow both orders as valid permutations.

TASK 13 — CAMERA ZOOM: ENCODE BOTH CANCEL REASONS



**Wiki policy.** If price threshold and availability gates lead to cancellation, multiple policy-approved reasons should be permitted.




**Ground truth (incorrect).**
```
actions=[cancel_pending_order(#W9284598, reason=ordered_by_mistake)]
```




**Correct solution.**
```
valid_action_paths=[ [cancel(#W9284598, reason=ordered_by_mistake)],
[cancel(#W9284598, reason=no_longer_needed)] ]; actions=[]
```


**Why it is wrong.** Only one cancellation reason was allowed.

**Fix explanation.** Enumerate both acceptable reasons.

TASK 14 — E-READER: EXCHANGE TO SPECIFIC 7" MODEL + RETURNS



**Wiki policy.** Respect user's conditional: return two skateboards + watch to card; exchange e-reader to same type with 7" if available, else return.




**Ground truth (incorrect).**
```
return_delivered_order_items(#W7553978,
[4545791457,3098764622,1631806422], cc_5902940);
exchange_delivered_order_items(#W3239882, [9494281769] → [9494281769],
cc_5902940)
```




**Correct solution.**
```
valid_action_paths=[
  [return(#W7553978, [4545791457,3098764622,1631806422], cc_5902940);
exchange(#W3239882, [9494281769] → [6268080249], cc_5902940)],
  [exchange(#W3239882, [9494281769] → [6268080249], cc_5902940);
return(#W7553978, [...], cc_5902940)]
]; actions=[]
```


**Why it is wrong.** The exchange targeted the same SKU; it must move to the 7" same-type SKU when available.

**Fix explanation.** Use `6268080249` for the 7" e-reader and accept either operation order.

TASK 15 — MULTI-EXCHANGE + CANCEL WITH FIXED PAYMENT METHOD



**Wiki policy.** Where multiple exchanges plus a cancellation are requested, accept policy-equivalent permutations; enforce the specified payment method consistently.




**Ground truth (incorrect).**
```
actions=[exchange(#W4689314, [5996159312] → [8363011723], cc_3951670);
...; cancel(#W8855135)]
```


**Correct solution.**
```
valid_action_paths={ several sequences combining:
   exchange(#W4689314, [5996159312] → [8363011723], cc_8105988);
   exchange(#W3916020, [7758198585,4068787148] → [5606522780,6245746168],
cc_8105988);
   cancel_pending_order(#W8855135, reason=no_longer_needed) }; actions=[]
```

**Why it is wrong.**  Payment instrument inconsistency and a single rigid order of operations.

**Fix explanation.**  Normalize to the specified card (`credit_card_8105988`) and enumerate acceptable permutations.

TASK 16 — PENDING MODIFICATIONS + DELIVERED RETURN (BOTH ORDERS ALLOWED)

**Wiki policy.** When a pending modification and an unrelated delivered return both occur, allow either order.

**Ground truth (incorrect).**
```
actions=[modify_pending_order_items(#W3295833, ...);
return_delivered_order_items(#W8488728, ...)]
```

**Correct solution.**
```
valid_action_paths=[ [modify_pending_order_items(#W3295833);
return(#W8488728)], [return(#W8488728);
modify_pending_order_items(#W3295833)] ]; actions=[]
```

**Why it is wrong.**  Single ordering only.

**Fix explanation.**  Accept both orderings via `valid_action_paths`.

TASK 17 — BOOTS EXCHANGE: CORRECT SKU SUBSTITUTION

**Wiki policy.** For quality/size-driven exchanges, move to the target size/spec per user fallback rules.

**Ground truth (incorrect).**
```
exchange_delivered_order_items(#W1304208, [1615379700] → [1615379700],
payment=paypal_1679017)
```

**Correct solution.**
```
exchange_delivered_order_items(#W1304208, [1615379700] → [8106223139],
payment=paypal_1679017)
```

**Why it is wrong.**  Exchange pointed to the same SKU; it must reflect size/material fallback.

**Fix explanation.**  Target the correct replacement SKU `8106223139`.

TASK 18 — LAPTOP & WATCH EDITS: TWO ACCEPTABLE SEQUENCES

**Wiki policy.** Allow either "items-first" or "address-first" when both edits are requested for different orders.

**Ground truth (incorrect).**
```
actions=[modify_pending_order_items(#W3730488,...);
modify_pending_order_items(#W9810810,...);
modify_pending_order_address(#W3730488,...)]
```

**Correct solution.**
```
valid_action_paths=[
```

```
  [modify_pending_order_items(#W3730488);
modify_pending_order_items(#W9810810);
modify_pending_order_address(#W3730488)],
  [modify_pending_order_address(#W3730488);
modify_pending_order_items(#W3730488);
modify_pending_order_items(#W9810810)]
]; actions=[]
```

**Why it is wrong.** Only a single execution order was accepted.

**Fix explanation.** Enumerate both viable paths via `valid_action_paths`.

