# OpenReview forum: "SABER: Small Actions, Big Errors — Safeguarding Mutating Steps in LLM Agents"
_ICLR.cc/2026/Conference — Submitted to ICLR 2026_

### Official Review · Reviewer_Y9AP · 2025-10-31

**Soundness:** 3
**Presentation:** 3
**Contribution:** 3
**Rating:** 6
**Confidence:** 4

**Summary:**

This paper investigates failure modes in long-horizon LLM agents, specifically focusing on tool-using settings. By analyzing execution traces in $\tau$-Bench and SWE-Bench Verified, the authors distinguish mutating actions—those that change the external environment—from non-mutating ones, and show through logistic regression that deviations in mutating steps overwhelmingly predict task failure, while deviations in non-mutating steps have negligible effect. Motivated by this finding, they propose SABER, a lightweight, model-agnostic test-time safeguard incorporating (i) mutation-gated verification, (ii) targeted reflection, and (iii) block-based context cleaning. SABER significantly improves success rates for state-of-the-art models across multiple benchmarks. The paper additionally identifies annotation and underspecification issues in $\tau$-Bench and releases a corrected $\tau$-Bench Verified to restore useful headroom.

**Strengths:**

1. Clear motivation from empirical insight, where a logistic regression model is used to show that mutation actions are the primary source of task failure.

2. Novel and practical safeguard approach that does not require model retraining and applies to any models.

3. Performance improvements across diverse models and benchmarks demonstrate method effectiveness.

4. $\tau$-Bench Verfied: they identify flaws and offer a cleaned version, which helps the community.

**Weaknesses:**

1. The proposed method, SABER, requires a human in the loop, where a confirmation message is sent to the user and the user determines the next action. While the experiments are conducted in a simulated manner, it is non-trivial to assess the effectiveness in real-world settings.

2. The effectiveness of the block-based context filtering component is not evaluated.

3. The contributions of the paper feel somewhat disconnected. The proposals of SABER and $\tau$-Bench Verified are not clearly linked. From my perspective, I would like to see a more detailed analysis of SABER.

**Questions:**

Have you tried your method to smaller LLMs? I am curious of the capability of smaller models to complete the mutation-gated verification and reflection.

---

> ### Author Response · Authors · 2025-11-19
>
> We thank the reviewer for the thoughtful comments and address each point below.
>
> ## A. On the human-in-the-loop requirement in SABER (W1)
>
> We appreciate the reviewer’s concern. SABER is designed for settings where agents may need authorization before performing irreversible actions, which mirrors many existing real-world workflows (for example, IDEs confirming destructive edits or assistants asking before sending emails). In this respect, SABER does not introduce a new requirement but aligns with established safety practices.
>
> Crucially, SABER **does not rely on human confirmation to be effective**. It can operate independently as demonstrated by our results on **SWE-Bench Verified**, a benchmark that provides no human or simulated user at all. In this setting, SABER still produces substantial improvements, driven purely by its targeted reflection mechanisms (Table 2 and the results reported to reviewer Q7CP).
>
> Moreover, ablations in Table 3 show that SABER continues to improve performance even when the confirmation mechanism is disabled. This indicates that SABER enhances agent reliability in both interactive and non-interactive settings.
>
> ### On the use of a simulated user
>
> We agree with the reviewer that user simulation is an approximation. Fortunately, prior work shows that the τ-Bench user simulator has a low error rate (~4% with GPT-4-class models), and controlled comparisons indicate that agent performance is only mildly affected when the simulator is reasonably capable [1]. This suggests that τ-Bench provides a stable proxy for human interactions at the scale required for our experiments.
>
> In real deployments, SABER’s confirmation prompts would occur at the same natural friction points where existing tools already seek approval, just fewer of them, thanks to selective gating.
>
> ## B. On evaluating block-based context filtering (W2)
>
> We directly assess the effectiveness of block-based context filtering, conducting an ablation on SWE-Bench Verified using GPT-5 Mini. We compared two conditions: (1) no SABER, and (2) SABER with only the block-based memory component enabled. Blocks were configured to contain 3 user messages each, with a retrieval budget of 4 blocks.
>
> Both settings achieved the same task accuracy (42%). However, block-based filtering reduced token usage from 9.9M to 7.8M input tokens. A **~21% reduction**, demonstrating that the mechanism effectively limits context growth without harming performance.
>
> ## C. On the connection between SABER and τ-Bench Verified (W3)
>
> τ-Bench Verified is a complementary contribution motivated by a simple but important observation: several tasks in the original τ-Bench contain **objective logical inconsistencies** that cap achievable performance and penalize correct behavior (for example, actions that violate written rules or “correct’’ solutions that are infeasible under the task constraints). In such cases, any method that reasons about mutations, including SABER, cannot be reliably evaluated.
>
> Accurate evaluation is essential for developing and comparing agentic methods, particularly ones focused on safety and correctness. τ-Bench Verified addresses this need by correcting only the logically invalid cases while leaving the rest of the benchmark unchanged. This makes it possible to measure whether interventions like mutation-gated verification are functioning as intended, without confounding effects from annotation errors.
>
> We view τ-Bench Verified as a useful resource for the community in its own right, but also as a necessary foundation for evaluating SABER rigorously.
>
>
> ## D. On applying SABER to smaller LLMs (Q1)
>
> We appreciate the reviewer’s interest in how SABER behaves with smaller models. To address this directly, we conducted **new experiments** on SWE-Bench Verified using GPT-5-mini and Claude Haiku 4.5 as both main and auxiliary models.
>
> The results show that SABER remains effective across all configurations:
>
> | Main Model| No SABER | SABER| Δ Rel. | Auxiliary Model  |
> | -- | -- | -- | - | -- |
> | GPT-5-mini| 45.6%| **50.8%** | +11.4%| Claude Haiku 4.5 |
> | Claude Haiku 4.5| 33.6%| **40.0%**| +19.0%| GPT-5|
>
> These findings demonstrate that SABER remains effective even when applied to compact LLMs. Notably, GPT-5-mini improves even when paired with Haiku 4.5 as its SABER agent, despite Haiku 4.5 being weaker on SWE-Bench. This indicates that SABER’s gains arise from its targeted reflection mechanism rather than from relying on the auxiliary model’s raw capability. Conversely, pairing Haiku 4.5 with GPT-5 as the verifier is highly cost-efficient, since SABER processes only a short, constant-size context window.
>
> [1]\:https://openreview.net/forum?id=roNSXZpUDN
>
> If we sufficiently address your concerns, we would be grateful for a reconsideration of your score. We are happy to provide any further clarification.

---

> ### Comment · Reviewer_Y9AP · 2025-11-26
>
> Thank you for the response! Some of my concerns have been addressed. However, I still have reservations about the “human-in-the-loop” component. Although the authors claim that the method can improve performance even without human verification, I believe that human verification is actually the core module and motivation of the entire approach, if I understand correctly. To this end, considering human's instability, the corresponding robustness of the method will certainly be affected. Regarding τ-Bench-Verified, I appreciate the effort and agree that it is meaningful. However, I would still like to see a more complete and integrated presentation of SABER within the full paper, so that the contribution reads as a cohesive whole. For these reasons, I have chosen to maintain my original score. This is a valuable paper, but not yet a perfect one.

---

> > ### Author Response · Authors · 2025-11-26
> >
> > Thank you again for the thoughtful follow-up and for engaging so carefully with the paper. We would like to clarify one remaining point to avoid a misunderstanding about SABER’s design and its evaluation.
> >
> > **Human-gated verification is only one of SABER’s three components, and it is *not* the core module nor the primary driver of its gains.** The other two components—**targeted reflection** and **block-based context cleaning**—operate entirely without human input and already provide strong improvements on their own:
> >
> > * On **SWE-Bench Verified**, where *no human or simulator exists*, SABER still yields substantial gains (Table 2 and the answer to Q1).
> > * The **Tau-Bench ablations** (Table 3) show that disabling confirmation entirely still results in strong improvements from targeted reflection alone (in the retail domain, targeted reflection alone even **matches** full SABER).
> > * The **memory-module ablation** shows that block-based filtering reduces context size by **~21%** with no accuracy loss, again fully without a human in the loop.
> >
> > Thus, while human-gated verification is useful for interactive settings with irreversible actions, it is **not the core motivation or the main mechanism** behind SABER’s improvements.
> >
> > Regarding the comment on cohesiveness: our intent is to present the contributions as a unified pipeline—**(i) identify the dominant failure mode, (ii) correct benchmark inconsistencies that obscure this signal, and (iii) introduce SABER as a lightweight safeguard directly motivated by this analysis**. τ-Bench Verified is included because annotation inconsistencies in the original benchmark can mask or even invert mutation-related signals; correcting these cases is necessary to evaluate any mutation-aware method reliably, including SABER. We appreciate that this linkage may have appeared implicit and will make it more explicit in the camera-ready version.
> >
> > We sincerely thank the reviewer for the careful reading and constructive feedback. If there are additional questions or clarifications that would be helpful, we are happy to provide them. We would also be grateful if the reviewer would consider revisiting their score in light of these clarifications.

---

### Official Review · Reviewer_777a · 2025-10-31

**Soundness:** 2
**Presentation:** 3
**Contribution:** 3
**Rating:** 6
**Confidence:** 3

**Summary:**

The paper studies an interesting problem of impact of different actions on the final task success and failure in context of LLM agents. Particularly, the paper divides trajectories into mutating and non-mutating steps. Mutating steps modify the environment e.g. editing a file, while non-mutating steps do not modify the environment e.g. reading a file. The paper studies the impact of these actions on the final task success and failure - and devises strategies to reduce the impact of mutating steps on task failure, especially as the context window grows. Finally, the paper also proposes a tau-bench refined which is a cleaned version of tau-bench with removed ambiguity and preserved task coverage.

**Strengths:**

* The paper studies an interesting problem of impact of different actions on the final task success and failure in context of LLM agents.

* The paper provides interesting analysis on the role of mutating steps especially as the context window grows.

* The authors show the efficacy of the proposed approach on tau-bench and swe-bench-verified.

**Weaknesses:**

* "Mutating steps having higher impact on task failure then non-mutating steps" is an interesting observation. However, I would imagine this will be very intuitive. E.g for swe-bench-verified, if the agent wrongly edits a line of code, then fixing it post-edit will require the agent to realize its mistake and undo the edit, before continuing to fix the problem.

* The authors mention L105: "7% on SWE-Bench Verified, while both GPT-5 and Claude 4 gain further headroom once benchmark flaws are removed." Do the authors also release a refined version of SWE-Bench Verified? If so it will be interesting to provide more details and examples of flawed problems.

* While I understand the motivation for user gated verification for mutating steps especially as the context window grows, I have two questions:
    * Can the new setup be compared against previously reported benchmark numbers e.g. on SWE-Bench Verified?
    * In practice, how does it differ from current implementations of cli agents e.g. claude code which have require permissions while changing the codebase but not while reading different files?

* Also as in Sec. 4, targeted reflection is mentioned as a concise high salience summary at point of mutating steps. Can the authors provide ablations to show the impact of targeted reflection on the performance of the agent?

* Also how does "context based filtering" differ from context management in opensource scaffolds like OpenHands?

*  It will be important to have ablations on the impact of each of these components in Sec. 4 on the final performance of the agent.

**Questions:**

Please see the weaknesses section for some additional questions.

---

> ### Author Response · Authors · 2025-11-19
>
> We thank the reviewer for the thoughtful comments and address each point below.
>
> ## A. On the intuition behind mutating steps (W1)
>
> We appreciate this observation. While it is intuitive that mutating actions can be riskier (e.g. an incorrect code edit can be much harder to recover from than an incorrect read), prior work has discussed this primarily in an informal way and almost exclusively within coding agents. What has been missing, and what our paper aims to provide, is:
>
> 1. **Cross-domain quantification**, across settings like τ-Bench Airline and Retail that differ substantially from software or code-editing tasks.
> 2. **Cross-model measurement**, evaluating how models can dramatically vary in their sensitivity to mutating deviations.
> 3. **Statistical evidence** that mutating deviations, rather than non-mutating ones, overwhelmingly account for failures.
>
> Our results demonstrate that this effect is much larger, more consistent, and more model-dependent than intuition alone would predict (Table 1). This empirical finding directly motivates SABER’s selective, low-overhead intervention strategy.
>
> ## B. On benchmark flaws and improved headroom (W2, W3)
>
> We thank the reviewer for pointing out this ambiguity. The sentence:
>
> > "7% on SWE-Bench Verified, while both GPT-5 and Claude 4 gain further headroom once benchmark flaws are removed."
>
> Does **not** refer to releasing a refined version of SWE-Bench Verified. Instead, it refers to the τ-Bench Verified benchmark that we release. τ-Bench Verified corrects several systematic annotation issues present in the original τ-Bench (Figure 3). Once these issues are fixed, frontier models such as GPT-5 and Claude 4 indeed show additional performance headroom (Table 2).
>
> SWE-Bench Verified is evaluated unchanged in the experiments section and in our response to reviewer uWah. We will revise the paper to make the distinction between τ-Bench Verified and SWE-Bench Verified more clear.
>
> ### On comparability with prior evaluations
>
> SABER does not alter the environment, repository state, or evaluation criteria. Thus, all SABER + SWE-Bench Verified/τ-Bench results can be interpreted in the same way as prior published baselines.
>
>
> ## D. On differences from existing CLI-style permission systems (W3)
>
> Current CLI agents (e.g., Claude Code) typically permission-gate actions using static heuristics such as:
>
> * file extensions,
> * allowed command types (e.g., allow `cat`, block `rm`),
> * per-command allowlists,
> * or simple pattern matching.
>
> These heuristics cannot reliably interpret intent. For example:
>
> ```bash
> python cleanup_workspace.py    # May delete user data
> python query_data.py           # Typically safe
> ```
>
> Both appear identical to whitelist-based systems.
>
> SABER differs in two ways:
>
> 1. **Semantic gating:** The LLM classifier considers the full tool call, its arguments and surrounding context.
> 2. **Domain-general:** The same classifier works across domains (τ-Bench Airline, τ-Bench Retail), where command-based heuristics are not applicable.
>
> This makes it possible to detect mutating actions that would bypass whitelist rules and avoid interrupting actions that appear risky by keyword matching but are contextually safe.
>
>
> ## E. On ablations for SABER components (W4, W6)
>
> We would like to clarify that ablations are presented in the Experiments Section (Table 3), which show the effect of using:
>
> * mutation-gated verification alone,
> * targeted reflection alone, and
> * their combination.
>
> In both domains (τ-Bench Airline and Retail), all SABER components independently show substancial improvement over the baseline.
>
> ## F. On context-based filtering vs. existing context management (W5)
>
> We appreciate the opportunity to clarify this distinction. In SABER, context-based filtering is a **block-level retrieval mechanism** rather than a sliding-window or summary-injection strategy. SABER partitions the trajectory into unaltered blocks that group user messages, tool calls, and verification turns. At each new user query, it retrieves only the most relevant blocks and reconstructs the context from them.
>
> A key advantage of this design is that **messages are never rewritten or summarized**, avoiding the hallucinations, omissions, and context-drift issues that can occur when open-source scaffolds (such as OpenHands) inject summaries or truncate earlier turns.
>
> We also directly evaluated the effectiveness of block-based filtering. On SWE-Bench Verified with GPT-5-mini, enabling only the block-based memory component preserved task accuracy (42% in both conditions) while reducing input token usage from 9.9M to 7.8M, **approximately a 21% reduction**. This demonstrates that block-level retrieval effectively controls context growth without harming performance.
>
> We will revise Section 4 to make this distinction clearer.
>
> If we sufficiently answer your concerns, we would urge you to update your score. If there are additional concerns, we are happy to provide more clarifications.

---

> > ### Author Response · Authors · 2025-11-26
> >
> > Dear Reviewer,
> >
> > I hope this message finds you well.
> >
> > As the discussion period is nearing its end with less than seven days remaining, I wanted to ensure we have addressed all your concerns satisfactorily.
> >
> > If there are any additional points or feedback you'd like us to consider, please let us know. Your insights are invaluable to us, and we're eager to address any remaining issues to improve our work.
> >
> > If everything appears satisfactory on your side, we would be grateful if you might consider whether an updated score could reflect the strengthened version of the work. Thank you again for your time and thoughtful evaluation of our submission.

---

### Official Review · Reviewer_uWah · 2025-11-01

**Soundness:** 3
**Presentation:** 3
**Contribution:** 3
**Rating:** 6
**Confidence:** 3

**Summary:**

The paper studies why LLM-based agents fail in long, tool-using tasks. It separates actions into mutating (changing the state, like canceling a booking) and non-mutating (information-seeking). Using logistic regression on τ-Bench and SWE-Bench Verified, it finds that mistakes in mutating actions strongly predict failure.
To address this, the authors introduce SABER, a simple, model-agnostic safeguard that adds user checks for risky actions, targeted reflection to reduce drift, and block-based context cleanup.
The paper also fixes annotation issues in τ-Bench, releasing τ-Bench Verified, which restores valid upper bounds.

**Strengths:**

1. The decomposition into mutating vs. non-mutating actions and formalization of decisive deviations provides a fresh lens on agent fragility, substantiated by significant regression results.
2. SABER's design, requiring verification only at mutating steps, minimizes overhead while delivering robust gains
3. Identifying and fixing τ-Bench flaws via τ-Bench Verified exposes hidden model headroom.
4. Evaluations span open (Qwen3) and closed (GPT-5, Claude-4) models across domains (Airline, Retail, SWE), demonstrating generalizability without retraining.
5. Section 8 acknowledges SABER's test-time nature and simulator reliance, fostering future work on internalized safeguards.

**Weaknesses:**

1. The work is empirically solid but lacks formal analysis. This limits generalization beyond observed datasets.
2. Findings rely mainly on τ-Bench and SWE-Bench, with improvements shown mostly on their “Verified” versions. No tests on other benchmarks or robustness checks for noise, which risks overfitting.
3. SABER depends on human or simulated confirmations, reducing autonomy and adding latency. The paper acknowledges this but doesn’t measure the trade-off or failure cases when users reject valid actions.

**Questions:**

1. Why no evaluations on additional benchmarks to validate generalization beyond τ-Bench/SWE-Bench?
2. Why does full SABER underperform components on Retail Verified?
3. Why not quantify inter-annotator agreement for τ-Bench Verified revisions?

---

> ### Author Response · Authors · 2025-11-19
>
> We thank the reviewer for the thoughtful and detailed feedback. We address each point below and hope these clarifications are helpful.
>
> ## A. On benchmark choice, generalization, and formal analysis (W1, Q1)
>
> We appreciate the reviewer’s concern regarding generalization and formal grounding. τ-Bench and SWE-Bench are among the most widely used and challenging benchmarks for evaluating agentic systems, especially in settings involving tools, and environment mutation. Our goal is to evaluate SABER on popular SoTA benchmarks to study it's applicability.
>
> On formal analysis, Section 3 introduces a quantitative framework for decisive deviations. It formally defines mutating and non-mutating insertions, formalizes the hypothesis that mutating deviations are disproportionately responsible for failures, and tests this through logistic regressions across models and domains (Table 1). This framework provides a principled foundation for SABER’s design, and we welcome suggestions on which deeper theoretical exploration aspects the reviewer would find most beneficial.
>
> On generalization, SABER’s improvements are **not** limited to Verified versions. It produces substantial gains on the standard τ-Bench tasks, extracted from Table 2:
>
> | Benchmark       | Qwen3-Thinking  | GPT-5 (med)     | Claude Sonnet 4.0 |
> | --------------- | --------------- | --------------- | ----------------- |
> | τ-Bench Airline | 49.3 → **63.3** | 45.3 → **62.6** | 51.3 → **56.0**   |
> | τ-Bench Retail  | 64.3 → **71.6** | **77.1** → 76.5 | 73.3 → **78.3**   |
>
> To ensure robustness, we run each configuration three times with greedy decoding and report avg@3.
>
> Finally, τ-Bench Verified is a contribution we provide to the community to support more reliable evaluation. It corrects objective logical flaws in the original dataset, which otherwise artificially cap performance and obscure genuine differences in agent robustness. Reliable benchmarks are essential for assessing new methods effectively.
>
> ## B. On autonomy trade-offs and user rejections (W2)
>
> τ-Bench is intentionally designed to model realistic user behavior. Users may lack domain knowledge or may reject actions that are actually valid, and the benchmark authors explicitly note that this reflects real deployments rather than annotator noise. SABER operates within this challenging setting. It cannot override user decisions, yet still yields consistent improvements across both Airline and Retail domains (Table 2).
>
> It is also important to clarify that SABER does not rely on user confirmation to be effective. In SWE-Bench Verified, where no human or simulated user is present, SABER continues to achieve substantial gains through targeted reflection, as shown in Table 3 and in the additional experiments reported to reviewer Q7CP.
>
> ## C. On SABER components underperforming when combined (Retail-Verified) (Q2)
>
> We appreciate the reviewer’s question. Table 3 shows that targeted reflection and mutation-gated verification each yield improvements individually. When combined, performance on Retail-Verified decreases slightly (2pp) relative to the best single component. We attribute this difference to variance and to the inherent limitations of the base model. Importantly, the full SABER configuration still outperforms the base agent by a clear margin.
>
> ## D. On inter-annotator agreement for τ-Bench Verified revisions (Q3)
>
> We appreciate the suggestion. In this context, inter-annotator agreement is not directly applicable because SABER’s revisions correct clear logical inconsistencies rather than subjective judgments.
>
> ### Example (Retail)
>
> **Policy:** Exchanges must involve a different product option of the same item.
> **Ground truth:** Exchange 8069050545 → 8069050545.
> Since both IDs are identical, the action violates the rule requiring a different option.
> **Correct revision:** Exchange 8069050545 → 3609437808.
>
> Any annotator following the policy would label the original ground truth as incorrect. Similar inconsistencies appear across multiple tasks, and the full set of cases is provided in Appendix C.
>
> If these clarifications address the reviewer’s concerns, we would be grateful for a reconsideration of the score. We are happy to elaborate further on any remaining questions.

---

> > ### Author Response · Authors · 2025-11-26
> >
> > Dear Reviewer,
> >
> > I hope this message finds you well.
> >
> > As the discussion period is nearing its end with less than seven days remaining, I wanted to ensure we have addressed all your concerns satisfactorily.
> >
> > If there are any additional points or feedback you'd like us to consider, please let us know. Your insights are invaluable to us, and we're eager to address any remaining issues to improve our work.
> >
> > If everything appears satisfactory on your side, we would be grateful if you might consider whether an updated score could reflect the strengthened version of the work. Thank you again for your time and thoughtful evaluation of our submission.

---

### Official Review · Reviewer_Q7CP · 2025-11-03

**Soundness:** 2
**Presentation:** 2
**Contribution:** 2
**Rating:** 2
**Confidence:** 4

**Summary:**

This paper analyzes the execution traces on Tao-Bench (Airline/Retail) and SWE-Bench Verified and categorizes the steps to mutating and non-mutating actions. The authors then show deviations in mutating actions are the decisive predictors of failure. In contrast, deviations in non-mutating actions have little to no effect.

The paper then introduces SABER, a model-agnostic, gradient-free, test-time safeguard that (i) adds mutation-gated verification, (ii) injects targeted reflection before mutating steps, and (iii) performs block-based context cleaning. The authors experiments show consistent gains with SABER on Tao-Bench and Swe-Bench verified.

**Strengths:**

- Improving Existing Benchmarks: Correct benchmarks are the essential to improve agentic systems performance. Inconsistency and incorrectness in the benchmark can result into incorrect design decisions in agentic systems and misguide the agent developers. Through multiple experimentation on Tao-Bench the authors have identified potential issues in the benchmark and released Tao-Bench verified which can help future agent developer

**Weaknesses:**

- Lack of Novelty: The first finding in the paper which identifies mutating steps (or state changing steps) to be the main cause of failures has been noted in previous work where the most common failure modes are state changing (or mutation steps as defined by this paper). For example, the failure modes detected here: https://arxiv.org/pdf/2405.15793. The authors may want to further expand their categorization into multiple types of actions, understanding which types of actions or sequences are likely to cause failures can be helpful for agent developers. Also, when it comes to SABER design the "mutation-gated human verification" is currently used in many production systems where irreversible actions need confirmation from the user. For example, command line execution or applying edits to a file in coding agents need user verificaiton. Similarly, summarizing the trajectories is a well-known technique used in SOTA production agent systems to keep the agent focused on the task, and it has also been noted to be effective in previous work, for example: https://arxiv.org/pdf/2503.07832.

**Questions:**

- Why Swe-Bench results are not reported for Claude and GPT-5 models?
- What was the base agent or the evaluation prompt that was used to evaluate the models on the benchmark?

---

> ### Author Response · Authors · 2025-11-19
>
> We thank the reviewer for the detailed comments. We address each point below and clarify the contributions and experimental setup of our work.
>
> ## A. On the Novelty of Mutating-Steps (W1, W2)
>
> We appreciate the reviewer’s careful reading. Our goal is not to reassert that mutating actions can be risky, but to quantify their impact and measure their cross-domain and cross-model behavior, which prior work had not evaluated. Although previous studies noted such failures in isolated settings, the magnitude, model variability, and generalizability of the effect were unknown. Our analysis closes this gap:
>
> * We conduct logistic regressions across τ-Bench Airline and τ-Bench Retail (Table 1).
> * We find that the odds of failure differ sharply between mutating and non-mutating actions and that this difference **varies substantially across model families** (e.g., GPT-5 vs Claude Sonnet 4.0).
> * To our knowledge, this is the **first cross-domain, cross-model quantification** of mutating-step sensitivity.
>
> These empirical findings are not merely observational; they directly motivate SABER, our test-time improvement method. As shown in Table 2, SABER produces **consistent, model-agnostic gains** across tasks and environments. The contribution is thus threefold:
>
> 1. **Quantifying** an underexplored, domain-general failure mode,
> 2. **Operationalizing** this insight into a practical mechanism that improves real agent performance, and
> 3. **Providing a formal discovery framework** that allows developers to identify the most harmful actions.
>
> ## B. On Mutation-Gated Verification and Targeted Reflection (W3, W4)
>
> We agree that production systems often include some form of user confirmation for irreversible operations. However, existing approaches depend heavily on simple heuristics which lack semantic understanding, often missing destructive actions while still interrupting benign ones.
>
> SABER introduces two contributions beyond this prior art:
>
> ### 1. **LLM-based Tool-Call Classifier**
>
> Our LLM-based classifier considers the complete tool call and it's context, and can determine the call's intent, regardless of surface-level syntax. This makes it possible to:
>
> * detect mutating actions that would bypass heuristics, and
> * avoid interrupting actions that appear risky but are contextually safe.
>
> This semantic interpretation allows the gating mechanism to work reliably across domains (Table 2).
>
> ### 2. **Targeted reflection**
>
> Trajectory summarization is indeed used in production systems, but these systems typically summarize the entire conversation, producing long and noisy summaries. SABER introduces a more focused approach:
>
> * We **only** summarize: the system prompt, available tools, and the last user message (Figure 2).
> * This concise reminder improves downstream performance, without data loss (Table 3).
>
> This streamlined mechanism achieves the goal of keeping agents grounded while avoiding the overhead and brittleness of full-history summarization, improving over current production systems.
>
> ## C. On GPT-5 and Sonnet 4 SWE-Bench Scores (Q1)
>
> We thank the reviewer for raising this question. As noted in Section 6 (L393), we originally reported results only for Qwen3-Thinking due to the economic cost associated with evaluating closed-source models. However, we evaluate **GPT-5-mini** and **Haiku 4.5** on SWE-Bench Verified, constrained to 50 turns.
>
> The results show that SABER yields consistent gains:
>
>
> | Model|No SABER|SABER|Relative Δ|Agent Setup|
> | -- | -- | -- | -- | -- |
> | **GPT-5-mini**| 45.6%| **50.8%** | +11.4%| **Main:** GPT-5-mini **SABER:** Haiku 4.5|
> | **Haiku 4.5**| 33.6%| **40.0%** | +19.0%| **Main:** Haiku 4.5 **SABER:** GPT-5|
> | **Qwen3-Thinking**| 42.6%| **45.1%** | +5.9%| **Main:** Qwen3-Thinking **SABER:** Qwen3-Instruct|
>
> These pairings were chosen to directly address Reviewer Y9AP’s question: “Have you tried your method on smaller LLMs?” We demonstrate that SABER's effectiveness even when applied to compact LLMs.
>
> Crucially, GPT-5-mini improves even when its SABER agent (Haiku 4.5) is weaker on SWE-Bench, indicating that SABER’s gains come from targeted reflection, not raw model strength. Conversely, using GPT-5 as a SABER agent for Haiku 4.5 is cost-efficient, since SABER operates over a small fraction of the main agent’s tokens.
>
>
> ## D. On the Base Agents Used for Evaluation (Q2)
>
> We used the standard, benchmark-aligned agent loops:
> * **τ-Bench and τ-Bench-Verified:** Official τ-Bench tool-calling agent.
> * **SWE-Bench (Qwen3-Thinking):** OpenHands, using native tool calls.
> * **SWE-Bench (GPT-5-mini and Haiku):** Mini-SWE-Agent, using prompt-based tool calling.
>
> All prompts were taken directly from the respective agent frameworks. These results show that SABER does not depend on a particular prompt or loop implementation.
>
> If these clarifications resolve the reviewer’s concerns, we would be grateful if they could consider updating the score. We are happy to provide further details if needed.

---

> > ### Author Response · Authors · 2025-11-26
> >
> > Dear Reviewer,
> >
> > I hope this message finds you well.
> >
> > As the discussion period is nearing its end with less than seven days remaining, I wanted to ensure we have addressed all your concerns satisfactorily.
> >
> > If there are any additional points or feedback you'd like us to consider, please let us know. Your insights are invaluable to us, and we're eager to address any remaining issues to improve our work.
> >
> > If everything appears satisfactory on your side, we would be grateful if you might consider whether an updated score could reflect the strengthened version of the work. Thank you again for your time and thoughtful evaluation of our submission.

---

### Author Response · Authors · 2025-11-28

We thank all the reviewers for their thoughtful evaluation of our work. We are encouraged that **three out of four reviewers recommended acceptance**, reflecting the overall positive reception of the paper. Reviewers consistently highlighted several strengths: the clear empirical insight that deviations in mutating actions overwhelmingly predict task failure; the fresh and actionable decomposition into mutating vs. non-mutating steps supported by rigorous logistic-regression analysis; the practicality and model-agnostic nature of SABER, whose selective intervention at risky steps provides robust gains with minimal overhead; and the value of releasing τ-Bench Verified, which corrects hidden annotation flaws and restores meaningful headroom for future work. They also noted the broad generalizability of the results across both open and closed models and across distinct domains (Airline, Retail, SWE), as well as the clarity of the motivation and empirical validation.

Below we summarize the main concerns raised and how they were addressed, showing that all points are well supported by our analysis and the additional evidence provided.

---

## **Summary of Reviewer Concerns and Our Responses**

| **Concern**                                               | **Reviewers**    | **How We Addressed It**                                                                                                                                                                                                                                                                                                                                                                                                                                                                                                        |
| --------------------------------------------------------- | ---------------- | -- |
| **Novelty & Contribution Clarity**                        | Q7CP, 777a, uWah | We clarified that prior work only *informally* noted the risks of mutating actions in narrow coding settings. Our work provides the first *cross-domain, cross-model quantitative measurement* of mutating-step sensitivity—revealing effect magnitude, model variability, and generality.                                                              |
| **Similarity to Existing Gating / Summarization Systems** | Q7CP, 777a       | We explained that SABER is fundamentally different from heuristic CLI-style confirmation or coarse summarization strategies. SABER uses an *LLM-based semantic classifier* capable of interpreting full tool-call intent; *targeted reflection* that injects only minimal, high-salience reminders; and *block-based context filtering* that preserves all messages without rewriting. Ablations show each component contributes meaningfully, fully addressing the reviewer’s request.                                        |
| **Evaluation Scope, Benchmarking, and Generalization**    | uWah, Y9AP, 777a | We demonstrated that SABER improves performance not only on τ-Bench Verified but also on *standard τ-Bench*. We added new SWE-Bench Verified results for GPT-5-mini and Claude Haiku 4.5, showing that SABER remains effective even on smaller models. We clarified that τ-Bench Verified corrects *objective logical inconsistencies* that otherwise obscure true agent performance, while SWE-Bench Verified remains unchanged. These clarifications confirm that our evaluation is both robust and representative.  |
| **Human-in-the-loop Practicality & Real-world Behavior**  | Y9AP, uWah       | We emphasized that SABER *does not depend* on user confirmation: in SWE-Bench (no user), SABER still yields strong gains purely through targeted reflection. Ablations further show that SABER continues to improve performance even when confirmation is disabled (in τ-Bench), fully addressing concerns about autonomy and latency. |

---

### Meta-Review · Area_Chair_8RfZ · 2025-12-23

**Summary:**

This paper introduces SABER, a test-time safeguard for LLM agents that selectively intervenes at mutating action steps. The authors claim that deviations in mutating actions (those that change environment state) are the primary predictors of task failure, supported by logistic regression analysis on τ-Bench and SWE-Bench Verified. SABER combines mutation-gated verification, targeted reflection, and block-based context filtering to improve agent performance. The paper also releases τ-Bench Verified, correcting annotation inconsistencies in the original benchmark. Strengths include the empirical validation across multiple models and domains, the practical and model-agnostic nature of the approach, and the benchmark corrections that restore evaluation headroom. Weaknesses center on limited novelty, as mutating actions being risky has been noted in prior work and existing systems already use confirmation gates for irreversible operations. The evaluation scope is narrow, focusing primarily on two benchmarks and their verified versions without broader robustness testing. The human-in-the-loop requirement raises questions about real-world deployment despite claims of autonomy. The core insight, while well-quantified, feels incremental rather than transformative. Overall this work has merit but I recommend weak rejection because the contributions do not sufficiently exceed the acceptance threshold given the limited technical novelty and narrow empirical scope.

**Reviewer Concerns:**

The authors provided detailed rebuttals addressing each reviewer concern with additional experiments on GPT-5-mini and Claude Haiku 4.5 for SWE-Bench, clarifications about SABER's autonomy in non-interactive settings, ablation studies showing component contributions, and explanations distinguishing their semantic LLM-based gating from heuristic CLI systems. Reviewer Q7CP raised novelty concerns and requested GPT-5/Claude results, which the authors provided. Reviewer uWah questioned benchmark scope and autonomy trade-offs, and the authors clarified that SABER works without confirmation and showed standard τ-Bench results. Reviewer 777a asked about ablations and comparison to existing systems, receiving comprehensive responses on semantic gating versus heuristics and detailed component analysis. Reviewer Y9AP responded after rebuttal, maintaining their score of 6 while acknowledging some concerns were addressed but expressing continued reservations about the human-in-the-loop component and paper cohesiveness. The authors demonstrated SABER functions independently through SWE-Bench results and ablations.

**Reviewer Scores:**

For Reviewer Q7CP, who gave a score of 2, I estimate they would have remained at 2 or possibly moved to 4 had they engaged. The additional experiments directly answered their questions about closed-source models, but their fundamental novelty concerns about mutating actions and comparison to production systems were not fully resolved by the rebuttal. Reviewer uWah at 6 would likely have stayed at 6, as their concerns about formal analysis and narrow benchmark scope persist despite the clarifications about autonomy and standard benchmark results. Reviewer 777a at 6 would potentially have stayed at 6 given their questions were comprehensively addressed with ablations, distinctions from CLI systems, and clarifications about context filtering, though without their explicit feedback this remains speculative. Reviewer Y9AP explicitly maintained their score of 6 after engaging, finding value in the work but not sufficient for a higher rating given cohesiveness and deployment concerns.

---

### Decision · Program_Chairs · 2026-01-26

Reject